# JHY enables the transition from switchable to fixed ciliary waveforms in metazoan evolution

Qingxia Chen[1,6], Shuxiang Ma [2,6], Hao Liu[3,6], Juyuan Liu[3], Qingchao Li [2], Qian Lyu[2], Hanxiao Yin [4], Junkui Zhao[4], Shanshan Nai[2], Ting Song[2], Hongbin Liu [5], Jun Zhou [2,4], Xiumin Yan [1], Xueliang Zhu [3✉] & Huijie Zhao [2✉]

## Abstract

**Motile cilia are evolutionarily conserved protrusions critical for motility and homeostasis. Their rhythmic movements require the central pair microtubules (CP-MTs). While the initial CP-MT assembly in mammals is mediated by WDR47 and microtubule minus-end-binding CAMSAPs, the mechanism by which CP-MTs are stabilized remains unclear. Here, we demonstrate that WDR47 coordinates JHY and SPEF1 to maintain the stability of mammalian CP-MTs. By generating a proximity interactome of WDR47, we identify a group of CP-MT-associated proteins, including SPEF1 and JHY. WDR47 enriches JHY and SPEF1 to the central lumen and tip of nascent cilia, whereas SPEF1 recruits WDR47 and JHY to CP-MTs through direct interactions. *Jhy* deficiency in mice preferentially disrupts distal CP-MTs, resulting in rotatory ciliary beats. Phylogenetic analyses suggest conserved functions of WDR47 and SPEF1 in protozoa and metazoans, as well as a role for JHY in animals with radial or bilateral body symmetry. We propose that JHY emerges to further reinforce CP-MTs, enabling the transition from switchable to fixed ciliary waveforms in metazoan evolution.**

**Keywords** Motile Cilia; Central Pair Microtubule; Ciliary Motility; Ciliopathy; Primary Ciliary Dyskinesia
**Subject Categories** Cell Adhesion, Polarity & Cytoskeleton; Evolution & Ecology; Organelles

## Introduction

Cilia are hair-like, microtubule (MT)-based organelles constructed on specialized centrioles called basal bodies. Motile cilia are widely present in protists, driving the active, rapid motility of these unicellular organisms in aqueous environments through their rhythmic beating. They are retained by the vast majority of metazoans in evolution. In mammals, for instance, motile cilia are distributed in epithelial cells lining brain ventricles, airways, and reproductive ducts as multicilia, i.e., multiple cilia per cell, and in sperm cells in the testis as monocilia (one cilium per cell), named flagella. While flagellar beats propel sperm movement to enable fertilization, multiciliary beats generate directional fluid flows over the epithelial surface to drive cerebrospinal fluid circulation, mucous clearance, and germ cell movement (Boutin and Kodjabachian, 2019; Derderian et al, 2023; Lyu et al, 2024; Zhu, 2025). Defects in multicilia cause primary ciliary dyskinesia (PCD), an inherited disorder with symptoms including recurrent airway infections, laterality disorders, infertility, and hydrocephalus in some cases (Wallmeier et al, 2020; Walton et al, 2023).

Typically, a motile cilium has a '9 + 2' axoneme arrangement, i.e., with the two central singlet MTs (C1 and C2) surrounded by nine peripheral doublet MTs that are decorated with multiprotein complexes, including axonemal dyneins and radial spokes (Chen et al, 2023; Gui et al, 2021b; Ishikawa, 2022; Leung et al, 2025; Meng et al, 2024; Walton et al, 2023). Furthermore, the CP-MTs are also densely associated with numerous multiprotein complexes, known as projections, forming the central apparatus (CA), which cooperates with radial spokes to regulate ciliary motility and waveform (Chen et al, 2023; Grossman-Haham et al, 2021; Gui et al, 2022; Han et al, 2022; Leung et al, 2025; Lyu et al, 2024; Meng et al, 2024; Samsel et al, 2021). Importantly, the CP-MT loss paralyzes *Chlamydomonas* flagella (Smith and Lefebvre, 1997) and alters the mammalian multiciliary beat pattern from whip-like to rotatory (Liu et al, 2021; Nozawa et al, 2013; Zheng et al, 2019).

Unlike peripheral doublet MTs, which are elongated from the basal body, CP-MTs are non-centrosomal MTs that require either nucleation or pre-existing seeds to form (Chen et al, 2025; Euteneuer and McIntosh, 1981; Smith and Lefebvre, 1997). Intriguingly, although the CA structure is highly conserved from unicellular protists to multicellular organisms (Carbajal-Gonzalez et al, 2013; Carvalho-Santos et al, 2011), CA proteins identified to be critical for CP-MT formation in protists do not appear to exhibit similar roles in mammals (Liu et al, 2021; Smith and Lefebvre,

[1]Ministry of Education-Shanghai Key Laboratory of Children's Environmental Health, Institute of Early Life Health, Xinhua Hospital, Shanghai Jiao Tong University School of Medicine, 200092 Shanghai, China. [2]Center for Cell Structure and Function, College of Life Sciences, Shandong Normal University, 250014 Jinan, China. [3]Key Laboratory of Multi-Cell Systems, Shanghai Institute of Biochemistry and Cell Biology, Center for Excellence in Molecular Cell Science, University of Chinese Academy of Sciences, Chinese Academy of Sciences, 200031 Shanghai, China. [4]State Key Laboratory of Medicinal Chemical Biology, Haihe Laboratory of Cell Ecosystem, College of Life Sciences, Nankai University, 300071 Tianjin, China. [5]Cheeloo College of Medicine, Shandong University, 250012 Jinan, China. [6]These authors contributed equally: Qingxia Chen, Shuxiang Ma, Hao Liu. ✉E-mail: xlzhu@sibcb.ac.cn; huijiezhao@sdnu.edu.cn

1997; Teves et al, 2014; Zhang et al, 2006; Zhang et al, 2007). Therefore, whether multicellular organisms have evolved distinct CP assembly mechanisms from unicellular ones is an outstanding question in cilia biology.

In addition to its crucial role in initiating CP-MT assembly, WDR47 also acts as a key regulator for CP-MT maintenance (Liu et al, 2021). Considering the highly dynamic nature of MTs, especially the tendency of MT plus ends to collapse rapidly (catastrophe) even in the presence of CAMSAPs (Hendershott and Vale, 2014; Jiang et al, 2014), how the CP-MT distal region is stabilized remains unclear. Interestingly, SPEF1 is a member of the end-binding (EB) protein family and functions by directly binding to CP-MTs (Legal et al, 2025; Zheng et al, 2019). Studies using *Juvenile hydrocephalus* (*Jhy*) transgenic mice also reveal an important role for JHY in the same process (Appelbe et al, 2013; Muniz-Talavera and Schmidt, 2017). However, whether their molecular functions are related to the WDR47-CAMSAPs axis remains unknown.

In this study, we identified a group of CP-MT-associated proteins, including SPEF1 and JHY, through a proximity ligation approach using WDR47 as the bait. We demonstrate that WDR47, SPEF1, and JHY collaborate through direct interactions to stabilize the CP-MTs, particularly in the distal region. Furthermore, our phylogenetic analyses suggest that WDR47 and SPEF1 are CP-MT regulators conserved from protozoa to mammals, whereas JHY initially emerges in Cnidarians, invertebrates with radial body symmetry during metazoan evolution, and is retained in bilaterians, animals with bilateral body symmetry, which consist of most invertebrates and all chordates. We propose that WDR47 and SPEF1 cooperate to stabilize the CP-MTs throughout evolution, whereas JHY further strengthens the CP-MTs to facilitate cilia formation with a fixed beat waveform seen in most animals.

# Results

## Acquisition of a motile cilia-related WDR47 interactome through proximity ligation

To gain insights into proteins functioning with WDR47 in CP-MT formation, we employed a proximity proteomics approach by fusing APEX2 to WDR47 to form a bait localizing in the ciliary central lumen (Fig. 1A) (Liu et al, 2021). APEX2 is an engineered ascorbate peroxidase that can biotinylate proteins in a small radius (less than 20 nm) in the presence of biotin-phenol and hydrogen peroxide ($H_2O_2$) (Rhee et al, 2013). Thus, the APEX2-WDR47 fusion would allow covalent biotinylation of closely spaced proteins (Fig. 1A). We expressed APEX2-WDR47 in multiciliated mouse ependymal cells (mEPCs) through adenovirus infection (Fig. 1A). Microscopic imaging revealed that biotinylation was specifically observed in cells treated with biotin-phenol, but not in cells treated with DMSO. In alignment with the ciliary localization of WDR47 (Liu et al, 2021), biotinylated proteins were also located in the central lumen and enriched at the ciliary tips (Fig. 1B), indicating that the labeling reaction occurs strictly in the vicinity of WDR47.

To identify the proximity proteins of WDR47, biotinylated proteins in mock-treated (control) or biotin-phenol and $H_2O_2$-treated (APEX2-WDR47) mEPCs expressing APEX2-WDR47 were purified using streptavidin-conjugated beads and subjected to shotgun mass spectrometry (MS). Using a cutoff of ≥twofold for the peptide ratio between APEX2-WDR47 and control samples, a total of 1923 proteins were identified (Dataset EV1). Among the candidate proteins, the known WDR47 interactors, calmodulin-regulated spectrin-associated proteins (CAMSAP1, CAMSAP2, and CAMSAP3), were conspicuously identified (Dataset EV1) (Buijs et al, 2021; Chen et al, 2020; Liu et al, 2021; Ren et al, 2022). Gene ontology (GO) enrichment analysis indicated that about 21% (410 proteins) of the total proteins were annotated to the 'cilium organization' term (GO: 0044782) (Fig. 1C). In addition, 189 proteins were enriched in the term that describes microtubule-based movement (GO: 0007018). Therefore, the identified proteins contain an interactome of WDR47, including proximity proteins in the central lumen of motile cilia.

## Identification of WDR47-related candidate CP-MT-associated proteins

Next, we tried to further narrow down the pool of proteins to WDR47-related candidate CP-MT-associated proteins. We reasoned that these proteins would exhibit markedly reduced abundance in *Wdr47* knockout (KO) ependymal cilia compared to wild-type (WT) cilia. As we have previously generated two independent datasets of quantitative MS-based proteomics data for purified WT and *Wdr47* KO ependymal cilia (Liu et al, 2021) (Data ref: Liu et al, 2021), we first used these data to identify differential hits through the following criteria: (i) the peptide number from a protein in the WT sample is more than 5, (ii) the ratio of peptide numbers for each protein in *Wdr47* KO and WT samples is less than 0.5, and (iii) the label-free quantification intensity ratio for each protein in *Wdr47* KO and WT samples is less than 0.2. Under these strict criteria, 116 and 90 proteins remained in the replicates, respectively (Fig. 1D; Datasets EV2 and EV3). We then analyzed the intersection among these two datasets and the interactome and identified 42 common proteins (Fig. 1D; Dataset EV4).

Among the 42 proteins were 19 known mammalian CP-MT-associated proteins (Fig. 1E). These included (i) CP-MT assembly regulators WDR47, CAMSAPs, SPEF1, and CCDC13 (Liu et al, 2021; Ren et al, 2022; Wu et al, 2025; Zheng et al, 2019); (ii) CA projection proteins SPAG17 (C1a-c-e), SPEF2 (C1b), ADGB (C1b), SPAG6 (C1), PCDP1 (C1), CCDC108 (C2a), HYDIN (C2b), SPAG16 (C1-C2 bridge), and KIF9 (Fang et al, 2024; Qu et al, 2023; Samsel et al, 2021); and (iii) proteins (LRGUK, CFAP69, DLEC1, ENO4, and MYCBPAP) whose homologs in *Chlamydomonas reinhardtii* and *Tetrahymena thermophila* have been located to CA projections through early genetic and biochemical researches (Cai et al, 2021; Dai et al, 2020; Fu et al, 2019; Hou et al, 2021; Joachimiak et al, 2021; Mitchell et al, 2005; Rao et al, 2024; Shamoto et al, 2018; Zhao et al, 2019b), as well as recent cryo-electron tomography and cryo-electron microscopy studies (Fu et al, 2019; Gui et al, 2022; Han et al, 2022). These results validate the quality of this group of candidates.

This group of candidates also included nine proteins localizing to other axonemal structures (Fig. 1E), including the inner dynein arm (IDA) component DNAH3, the modifier-of-inner-arms (MIA) complex component CFAP73 (Walton et al, 2023; Wang et al, 2024; Yamamoto et al, 2013), axonemal microtubule inner proteins (MIPs) EFCAB6 and EFHC2 (Gui et al, 2021a), radial spoke component RGS22 (Leung et al, 2025), and transition zone proteins

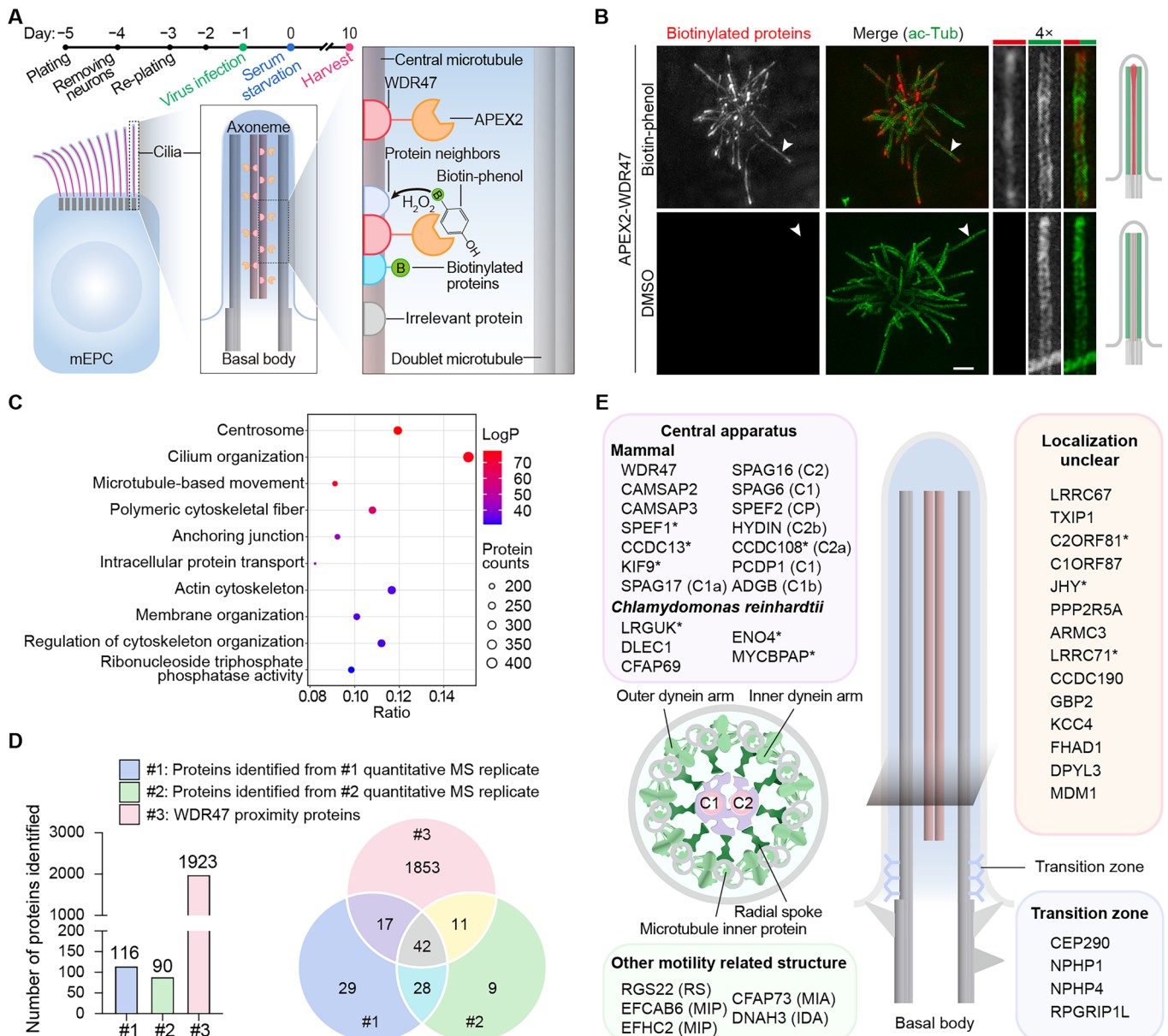

**Figure 1. Identification of WDR47-related candidate CP-MT-associated proteins.**

(**A**) Schematic diagrams of the APEX2-mediated proximity protein labeling strategy. Radial glial cells were cultured as illustrated to differentiate into multiciliated cells, and adenovirus or lentivirus was used to mediate the expression of exogenous proteins. The proximity proteins of WDR47 were biotinylated with 500 µM biotin-phenol by adding 1 mM $H_2O_2$. (**B**) Fluorescence images of mEPCs expressing APEX2-WDR47 treated with biotin-phenol or DMSO in the presence of 1 mM $H_2O_2$. Biotinylated proteins were visualized by Alexa Fluor 546-conjugated streptavidin, with acetylated α-tubulin (ac-Tub) labeling the cilium. Magnified images of individual cilium indicated by arrowheads were shown on the right. Diagrams show the specific proximity protein labeling in the central lumen. Scale bar, 2 µm. (**C**) Gene ontology (GO) enrichment analysis of WDR47 proximity proteins identified by mass spectrometry. The top 10 GO terms were presented. 833 proteins grouped into other terms were not shown. The hypergeometric test was used to determine the significance of the enrichment results. (**D**) Selective identification of CA-related proteins by integrating WDR47 proximity interactome with the quantitative proteomic profiles of *Wdr47* knockout motile cilia. The two replicates of quantitative proteomic analysis of *Wdr47* knockout motile cilia (Data ref: Liu et al, 2021) were used to identify differential hits. Forty-two candidate proteins were identified simultaneously in all three datasets. (**E**) Categories of proteins identified in (**D**). RS radial spoke, IDA inner dynein arm, MIP microtubule inner protein, MIA modifier-of-inner-arms complex, CP central pair microtubules C1 and C2. Proteins with their localization examined in this study were marked by an asterisk. Source data are available online for this figure.

CEP290, RPGRIP1L, NPHP1, and NPHP4 (Goncalves and Pelletier, 2017; Park and Leroux, 2022).

The remaining 14 candidates lacked localization information (Fig. 1E). Nevertheless, 6 proteins, including LRRC67 (Wang and Sperry, 2011), TXIP1 (Sultana et al, 2023), ARMC3 (Pausch et al,

2016; Rahim et al, 2024), GBP2 (Chen et al, 2017), KCC4 (Klein et al, 2006), and FHAD1 (Zhang et al, 2024), have been functionally linked to male fertility. Furthermore, one protein, JHY, has been implicated in CP-MT formation (Appelbe et al, 2013; Muniz-Talavera and Schmidt, 2017).

## JHY is a CP-MT-associated protein enriched in nascent short cilia like WDR47

Next, we examined subcellular localizations of selected candidate proteins with three-dimensional structured illumination microscopy (3D-SIM), which has a lateral optical resolution of ~120 nm. We infected cultured mEPCs with lentiviral particles, as illustrated in Fig. 1A, to express GFP-tagged proteins. As expected, GFP-CCDC108 indeed colocalized with HYDIN (Samsel et al, 2021) in the central lumen of ependymal motile cilia (Fig. 2A). GFP-tagged ENO4, LRGUK, and MYCBPAP also displayed a central lumen localization with HYDIN (Fig. 2A), confirming them as evolutionarily conserved CA proteins. Next, we examined the cellular distribution of several candidate proteins categorized as "Localization unclear". Immunofluorescence analysis revealed that GFP-tagged C2ORF81, LRRC71, CCDC13, and JHY co-localized with HYDIN in the central lumen (Fig. 2B), indicating their CA localization. Notably, GFP-JHY displayed pronounced accumulation both in the central lumen and at the ciliary tip in short cilia that were still negative for HYDIN (Fig. 2B), similar to WDR47 (Liu et al, 2021), suggesting an involvement of JHY in WDR47-mediated CP-MT formation.

As the diameter of the CA is approximately 50 nm, we reasoned that dual-channel stimulated emission depletion (STED) nanoscopy (Gottfert et al, 2013) would enable the identification of bona fide CP-MT-associated proteins. We thus employed a STEDYCON system, which achieves a lateral resolution of 30 nm, and first used the radial spoke head component RSPH4 (Curry et al, 1992; Meng et al, 2024; Yoke et al, 2020; Zheng et al, 2021) and the CP-MT-associated protein GFP-SPEF1 (Zheng et al, 2019) as controls. Due to severe fluorescence quenching, a single optical section was imaged for each microscopic field. Consistent with the arrangement of radial spoke heads closely surrounding the CA (Meng et al, 2024; Zheng et al, 2021), the immunofluorescence of RSPH4 was resolved as a ring of approximately 88 nm in diameter in optical cross-sections of axonemes visualized with acetylated tubulin (Fig. EV1A). In the central lumen of laterally oriented axonemes, a tube-like image of similar diameter was visualized for RSPH4 (Fig. EV1A). By contrast, GFP-SPEF1 in the central lumen appeared as a small punctum in optical cross-sections or numerous linearly aligned small puncta in laterally oriented axonemes (Fig. EV1B), indicating that the STED nanoscopy is sufficient to distinguish typical CP-MT-associated proteins from radial spoke components. Subsequent imaging indicated that GFP-tagged ENO4, C2ORF81, and JHY closely resembled the distribution patterns of GFP-SPEF1 (Fig. 2C), confirming them as typical CP-MT-associated proteins. In contrast, GFP-CCDC13 appeared to occupy a broader area than other analyzed CP-MT-associated proteins (Fig. 2C).

We then further clarified the localization of JHY with the STED nanoscopy. As the *Chlamydomonas* orthologues of KIF9 and ENO4, KLP1 and enolase, are projection components of C2 and C1 MTs, respectively (Gui et al, 2022; Han et al, 2022), we co-immunostained GFP-ENO4 with endogenous KIF9 and found that their CP localizations were clearly separated (Fig. 2D), consistent with the localizations of their *Chlamydomonas* counterparts. When GFP-JHY were co-immunostained with KIF9, their immunofluorescent signals were nicely aligned within the central lumen (Fig. 2D), suggesting that the localization of JHY is either associated with or close to the C2 MT.

## JHY colocalizes with WDR47 in motile cilia and interacts with WDR47

To exclude possible localization artifacts of exogenous JHY, we raised a rabbit antibody against a fragment of JHY (1–365 amino acid [aa]) and performed immunostaining. 3D-SIM indicated that, similar to GFP-JHY (Fig. 2B), endogenous JHY colocalized with HYDIN in the ciliary central lumen (Fig. 3A). Furthermore, it was also highly enriched at the tip of short cilia lacking the HYDIN staining (Fig. 3A).

Next, we examined the relationship between JHY and WDR47 in ependymal motile cilia. As both JHY and WDR47 antibodies are derived from rabbits, we examined mEPCs expressing GFP-JHY. In long (mature) cilia, JHY was evenly distributed along the CP-MT (Fig. 3B,C), whereas WDR47 showed a biased enrichment at the proximal CP-MT region (Fig. 3B,C), a pattern attributed to its associations with CAMSAPs (Liu et al, 2021; Ren et al, 2022). In short (nascent) cilia, WDR47 was intensely concentrated in the central lumen and ciliary tip prior to the emergence of JHY (Fig. 3B,C). When JHY accumulated in the central lumen and at the tip of slightly longer (growing) cilia, it colocalized with WDR47 (Fig. 3B,C). These results suggest that WDR47 enters nascent cilia to enrich JHY for the CP-MT formation.

Next, we clarified whether JHY and WDR47 could interact. Co-immunoprecipitation analysis using ectopically co-expressed GFP-JHY and Flag-WDR47 in HEK293T cells indeed revealed an interaction (Fig. 3D). To characterize the interaction in more detail, we mapped the binding sites between JHY and WDR47 (Fig. 3E–G). Although a conserved JHY domain exists in the C-terminus, the first 150 residues of JHY are necessary and sufficient to bind WDR47 (Fig. 3E,G). Similarly, by using various fragments of WDR47, we found that the C-terminal WD40 domain is required for binding to JHY (Fig. 3F,G), which has been reported to mediate the ciliary localization of WDR47 (Liu et al, 2021). Next, we performed a GST pull-down assay to verify whether JHY and WDR47 could directly interact. As shown in Fig. 3H, an interaction was detected using purified recombinant fragment proteins of WDR47 and JHY. These results suggest that JHY directly interacts with the C-terminal WD40 domain of WDR47 via its N-terminal region.

### *Jhy*-deficient mice develop PCD-related phenotypes

*Jhy^{lacZ/lacZ}* transgenic mice on the FVB/N background have been reported to develop rapidly progressing juvenile hydrocephalus (Appelbe et al, 2013). However, the underlying pathogenic mechanisms are yet to be explored. To characterize the expression pattern of JHY, we performed real-time polymerase chain reaction (PCR) analysis to determine the relative abundance of *Jhy* mRNA in a panel of mouse tissues. As shown in Fig. 4A, *Jhy* mRNA showed high expression levels in tissues abundant in motile cilia or flagella, an expression pattern consistent with its localization in motile cilia, indicative of a role of JHY in motile cilia or flagella.

Next, we generated a *Jhy* KO mouse model on the C57BL/6 J background using CRISPR-Cas9 with dual single guide RNAs (sgRNAs) (Fig. 4B). Genome deletions introduced by sgRNAs were confirmed by PCR (Fig. 4C). Immunostaining of WT and *Jhy* KO mEPCs further confirmed the loss of JHY in motile cilia and also validated the antibody specificity (Fig. 4D). *Jhy* KO mice were born at the expected Mendelian ratios and rapidly developed a noticeably domed head and hunched back, accompanied by growth retardation and early death within the first two postnatal weeks (Fig. 4E,F). Dissection of the *Jhy* KO brain revealed a greatly enlarged cerebrum

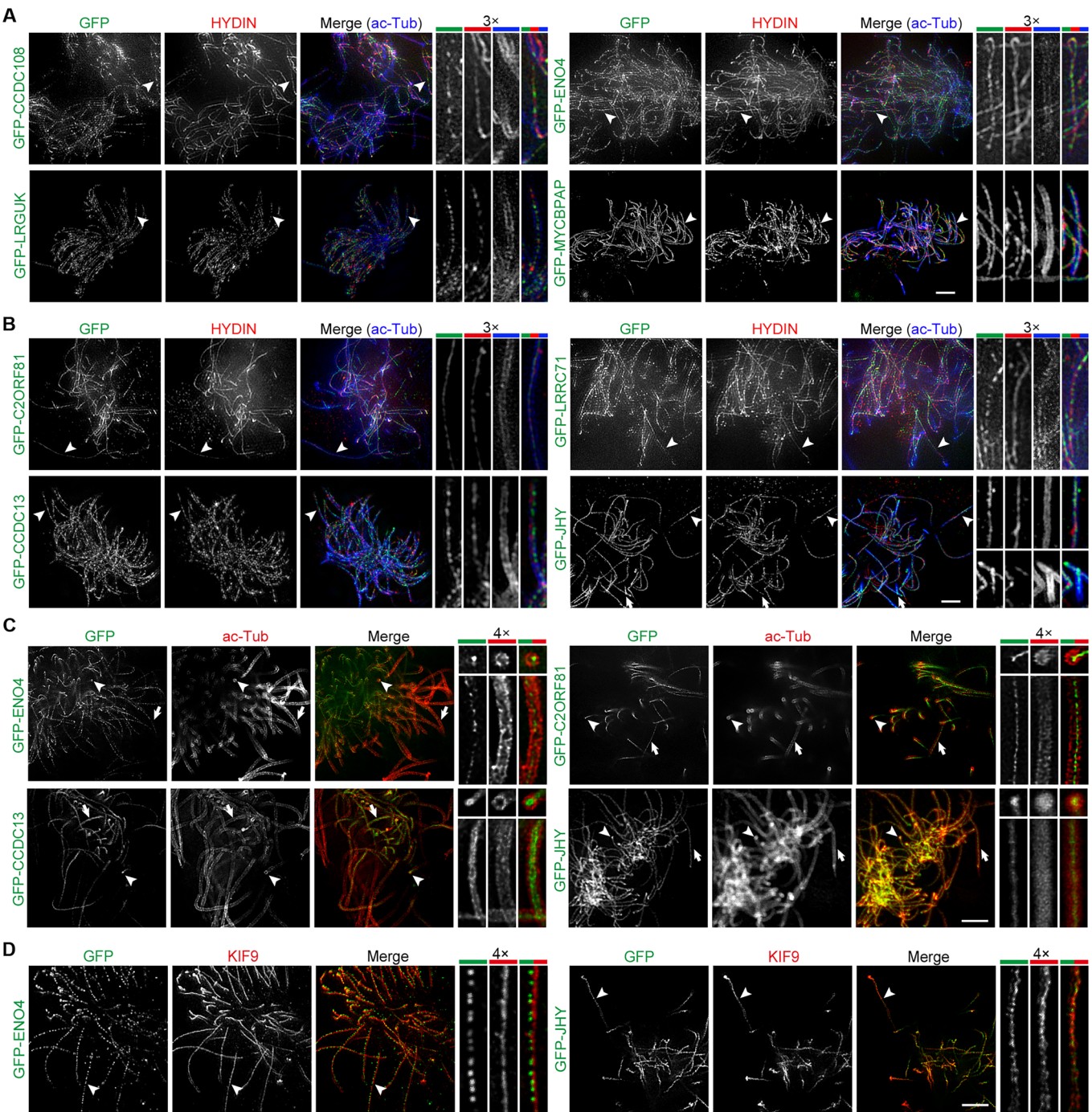

**Figure 2.   Validation of CP-MT-associated proteins with super-resolution microscopy.**

(A, B) 3D-SIM images of mEPCs exogenously expressing indicated GFP-tagged proteins. mEPCs were cultured as shown in Fig. 1A and labeled with antibodies against HYDIN and acetylated α-tubulin (ac-Tub) to indicate the CP and the axoneme of motile cilia. Magnified images of the mature long cilia and growing short cilia indicated by arrows and arrowheads were shown on the right. Note that GFP-JHY shows a substantial accumulation at the ciliary tip in growing short cilia. Scale bar, 2 μm. (C) STED images of mEPCs exogenously expressing indicated GFP-tagged proteins. Cells were serum-starved to induce multiciliogenesis and labeled with an acetylated α-tubulin (ac-Tub) antibody. Magnified images on the right show the longitudinal and transverse views, indicated by arrows and arrowheads, respectively. Note that GFP-CCDC13 displays a ring-like distribution in the central lumen. Scale bar, 2 μm. (D) STED images of mEPCs exogenously expressing indicated GFP-tagged proteins. Cells were serum-starved to induce multiciliogenesis and labeled with a KIF9 antibody. Magnified images of the region indicated by arrowheads were shown on the right. Scale bar, 2 μm. Source data are available online for this figure.

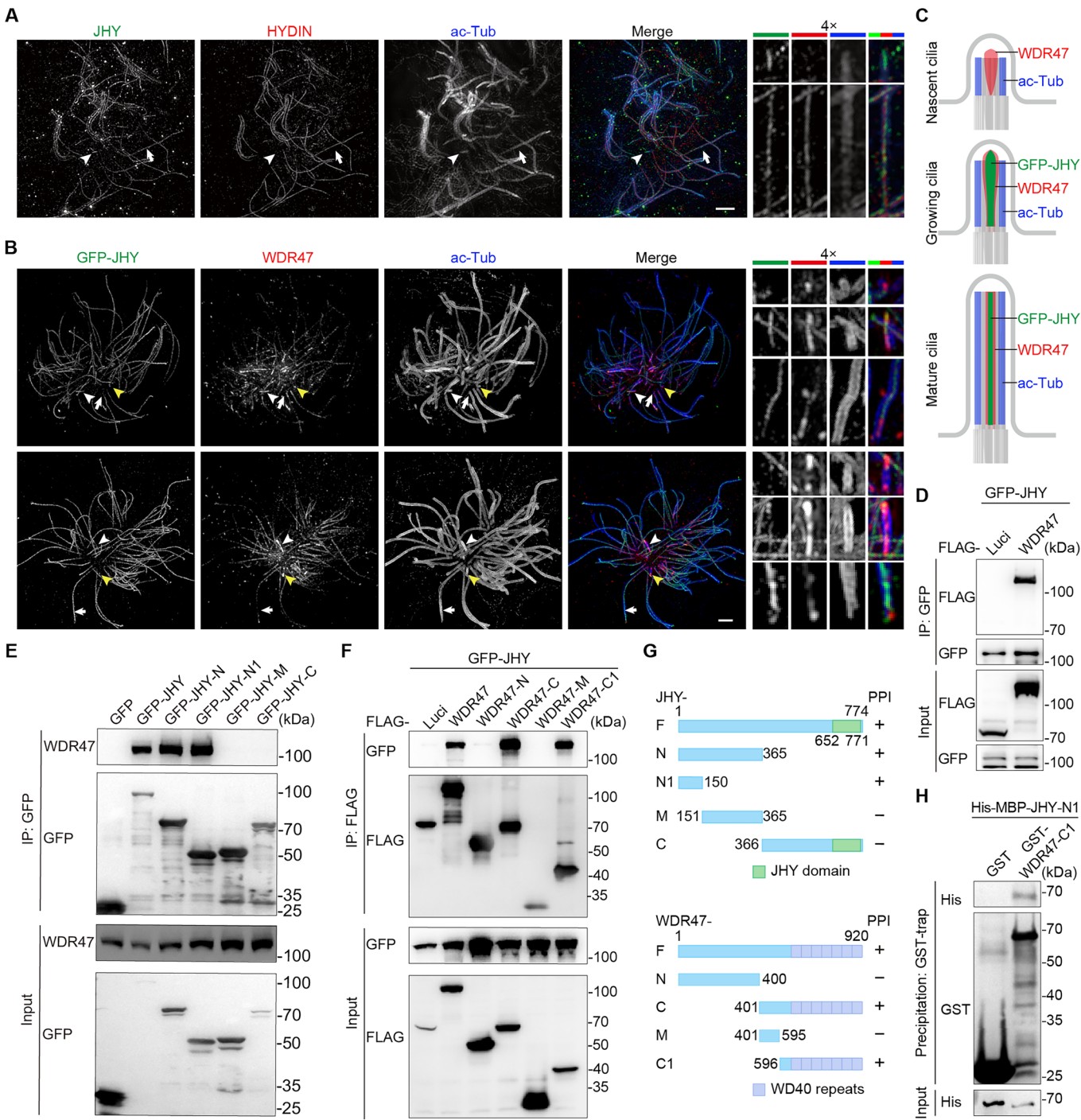

**Figure 3. JHY localizes to the central lumen and directly interacts with WDR47.**

(A) 3D-SIM images of mEPCs immunostained with the indicated antibodies. Magnified images of the mature long cilia and growing short cilia indicated by arrows and arrowheads were shown on the right. Note that JHY shows an accumulation at the ciliary tip in growing short cilia. Scale bar, 1 μm. (B) 3D-SIM images of mEPCs expressing GFP-JHY immunostained with the indicated antibodies. Nascent cilia, growing cilia, and mature long cilia were indicated by yellow arrowheads, white arrowheads, and white arrows, respectively. Note that WDR47 is located at the ciliary tip ahead of JHY. Scale bar, 2 μm. (C) Diagrams illustrate the spatial and temporal distributions of indicated proteins. (D–F) Co-immunoprecipitation (Co-IP) analysis in HEK293T cells exogenously expressing the indicated proteins. GFP-tagged proteins in (D, E) and Flag-tagged proteins in (F) were immunoprecipitated with anti-GFP and anti-Flag agarose beads, respectively. Blots were probed with the indicated antibodies. Luci luciferase. (G) Diagrams of JHY and WDR47 truncated fragments. Fragments were generated based on the domain prediction with SMART (a Simple Modular Architecture Research Tool). (H) GST pull-down assay using purified GST, GST-tagged WDR47 fragment, and His-MBP-tagged JHY fragment protein. Blots were probed with the indicated antibodies. Source data are available online for this figure.

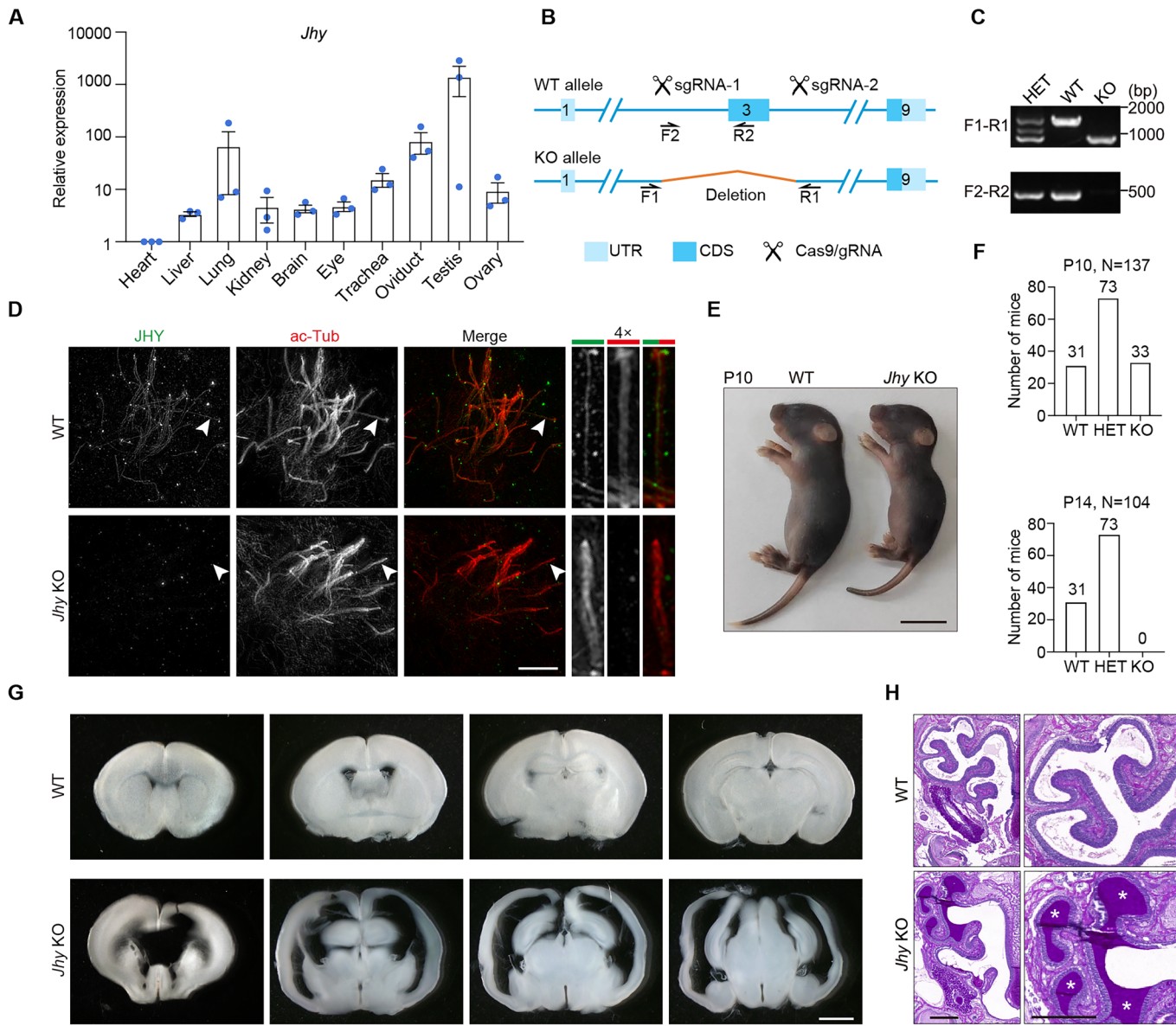

**Figure 4. *Jhy* knockout mice display PCD-related phenotypes.**

(A) Real-time PCR analysis of *Jhy* mRNA abundance in various mouse tissues. The expression of *Jhy* in various tissues was normalized using the corresponding *Gapdh* as the reference gene and baseline 1 (heart) as the reference sample ($\triangle\triangle C_T$ method). Data are from three independent biological repeats and presented on a logarithmic scale (log10) as mean ± SEM. (B) A schematic diagram of wild-type (WT) and *Jhy* knockout (KO) alleles. The genomic positions of primers (F1, R1, F2, and R2) used for genotyping were indicated. UTR, untranslated region; CDS, coding sequence. (C) Genotyping of WT, heterozygous (HET), and *Jhy* KO mice. (D) 3D-SIM images of WT and *Jhy* KO mEPCs immunostained with the indicated antibodies. Magnified images of individual cilia indicated by arrowheads were shown on the right. Note that the central lumen labeling of JHY in *Jhy* KO mEPCs is invisible. Scale bar, 5 μm. (E) Representative images of WT and *Jhy* KO mice at P10. Scale bar, 1 cm. (F) Quantifications of mice of the indicated genotypes at P10 and P14. Note that *Jhy* KO mice were born at Mendelian ratios but died within two weeks of birth. (G, H) Representative images of brain and nasal cavity sections of WT and *Jhy* KO mice. Brain sections of *Jhy* KO mice display enlarged ventricles and a dramatic decrease in cortical thickness. Histological staining of *Jhy* KO nasal cavities reveals abundant accumulation of protein-rich mucus. The asterisks indicate mucus accumulation. Scale bars, 2 mm (G) and 100 μm (H). Source data are available online for this figure.

filled with fluid, and serial sectioning confirmed the hydrocephalus condition with profoundly enlarged ventricles and a clear decrease in cortical thickness (Fig. 4G). Moreover, histological analysis of the sinus cavities showed abundant accumulation of protein-rich mucus in *Jhy* KO mice (Fig. 4H), indicating *Jhy* KO mice had severe sinusitis. Together, these results show that *Jhy* KO mice develop PCD-related phenotypes, including hydrocephalus and sinusitis.

## *Jhy* deficiency preferentially causes the loss of distal CP-MTs

Abnormal cilia structure or motility usually causes PCD-related phenotypes. To investigate the potential cellular mechanisms underlying these phenotypes in *Jhy* KO mice, we conducted scanning electron microscopy (SEM) to examine the ependyma

isolated from WT and *Jhy* KO mice. Compared to WT controls, the number of cilia appeared to decrease dramatically in *Jhy* KO tissue (Fig. 5A). In addition, the ciliary length appeared to be mostly uniform in WT samples but varied considerably in *Jhy* KO ependymal cells. To further verify the ciliary defects, we analyzed cilia formation in cultured mEPCs. Interestingly, there were no significant differences in the numbers of basal bodies and cilia between WT and *Jhy* KO mEPCs (Fig. 5B,C), suggesting that JHY is not essential for centriole amplification or cilia formation during multiciliogenesis. The ciliary defects observed in the ependyma of *Jhy* KO mice likely resulted from disruptions in epithelial integrity due to hydrocephalus.

Next, we examined ciliary motility in cultured mEPCs. Notably, while the WT motile cilia were observed to beat in a rhythmic whip-like manner, the *Jhy* KO motile cilia displayed a rotational movement with decreased frequency (Fig. 5D,E; Movie EV1). Given the necessity of CA for proper ciliary motility and the CA localization of JHY in motile cilia, we investigated whether the loss of JHY impairs the CA structure. Immunostaining of *Jhy* KO mEPCs revealed a complete loss of ciliary HYDIN in the motile cilia (Fig. 5F), indicating that *Jhy* deficiency severely disrupts the CA integrity. Furthermore, we performed transmission electron microscopy (TEM) to examine the ciliary ultrastructures. While WT motile cilia exhibited a normal "9 + 2" axonemal arrangement, most *Jhy* KO motile cilia had severe CP-loss defects, with one or both central MTs absent in the central lumen (Fig. 5G). When assessing the CP-loss defects in the proximal and distal regions of ciliary axonemes, distinguished by the presence or absence of microvilli around the axonemes, we found that a greater number of axonemes in the distal region tended to lack CP-MTs (Fig. 5H). Intriguingly, this observation differs from the CP-MT defects in *Wdr47* KO cilia, where all examined axonemes were completely devoid of both CP-MTs, suggesting a distinct role for JHY in this process.

## JHY interplays with WDR47 and SPEF1 to stabilize the CP-MTs

To clarify the relationship between JHY and WDR47, we examined the effects of JHY depletion on the ciliary localization of WDR47. In *Jhy*-deficient mEPCs, the distribution of WDR47 in the central lumen disappeared in most long (mature) cilia, only occasionally leaving some immunofluorescent signals (Fig. 6A–C), possibly due to remnant central MTs (Fig. 5G). In sharp contrast, WDR47 in short (growing) cilia remained unaffected (Fig. 6A–C). These results are consistent with the observation that WDR47 was enriched in nascent cilia earlier than JHY (Fig. 3B), thus confirming that WDR47 functions upstream of JHY.

Next, we investigated their functional mechanisms. As WDR47 is not an MT-binding protein (Buijs et al, 2021; Chen et al, 2020), we examined whether JHY could bind to MTs. GFP-JHY expressed in U2OS cells, however, displayed no signs of MT association and was mainly aggregated in the cytoplasm (Fig. 6D). As SPEF1 and CAMSAPs are known MT-binding proteins critical for CP-MT formation (Liu et al, 2021; Zheng et al, 2019), we respectively co-expressed SNAP-tagged SPEF1 and CAMSAP1 with GFP-JHY. Interestingly, JHY was relocated to the SPEF1-bundled MTs but not to MTs decorated by CAMSAP1 (Fig. 6D). Furthermore, when WDR47 was co-expressed with SPEF1, it was also recruited to

SPEF1-bundled MTs (Fig. 6E). Therefore, both JHY and WDR47 can be recruited to MTs by SPEF1.

Through co-immunoprecipitation analysis, we demonstrated that WDR47 interacted with SPEF1 via its C-terminal WD40 domain (Fig. 6F,G). In addition, we found that JHY also exhibited an interaction with SPEF1 (Fig. 6H). The interaction between JHY and SPEF1 required its C-terminal region, which contains the conserved JHY domain (Fig. 6I). Taken together, we conclude that SPEF1 recruits JHY and WDR47 to central MTs to maintain their stabilization.

## Conservation of the WDR47, SPEF1, and JHY in evolution

To investigate the origin and physiological significance of the WDR47-SPEF1-JHY axis in evolution, we conducted phylogenetic analyses. BLAST (Basic Local Alignment Search Tool) searches across representative taxa revealed that WDR47 and SPEF1 are conserved from protozoa to mammals in organisms with motile cilia or flagella (Fig. 7A–C). For WDR47, both the C-terminal WD40 repeats-containing region, capable of localizing to the ciliary central lumen and binding to both SPEF1 and JHY (Figs. 3F and 6G), and the N-terminal domain (1–315 aa), essential for CAMSAP interaction and dimer formation (Chen et al, 2020; Ren et al, 2022), are highly conserved, except that *Drosophila* WDR47 lacks the N-terminus (Fig. 7B) (see Discussion). Both the N-terminal MT-binding Calponin homology (CH) domain and the C-terminal dimerization domain of SPEF1 (Zheng et al, 2019) are also highly conserved (Fig. 7C), supporting their critical roles in the CP-MT formation. Consistently, SPEF1 homologs are absent in Rhabditida nematodes (Fig. 7A), including *Caenorhabditis elegans*, which lack motile cilia or flagella (Inglis et al, 2007). However, they are present in other nematode families, particularly parasitic nematodes (e.g., *Toxocara canis*; GenBank accession KHN77366), for unknown reasons. On the other hand, WDR47 homologs are widely present in Nematoda (Fig. 7A,B), probably due to their roles in neurons (Buijs et al, 2021; Chen et al, 2020).

In contrast, no JHY homologs were identified in protozoa, even when the JHY domain was used for BLAST searches. In metazoa, JHY homologs were absent from Placozoa, Porifera, and Cteno-phora, three of the earliest metazoan phyla. They were widely present in Cnidarians and bilaterians (Fig. 7A,D), which respec-tively have radial and bilateral body symmetries (Genikhovich and Technau, 2017; Senatore et al, 2016). These findings suggest that JHY may specifically emerge in Cnidarians and bilaterians to reinforce the CP-MT structure by interacting with WDR47 and SPEF1 in motile cilia (Figs. 3D and 6F).

## Discussion

The formation of a functional CA requires the assembly and stabilization of the CP-MTs as prerequisites. In this study, we demonstrate that WDR47 forms a complex with JHY and SPEF1 to stabilize CP-MTs (Fig. 7E). In mammals, WDR47 recruits CAMSAPs to the proximal ciliary central lumen to stabilize CP-MT seeds generated by the MT-severing protein KATANIN, facilitating CP-MT elongation (Chen et al, 2025; Liu et al, 2021). Here, our results show that WDR47 also recruits SPEF1 and JHY to the central lumen and tip of nascent cilia (Fig. 7F). Given that

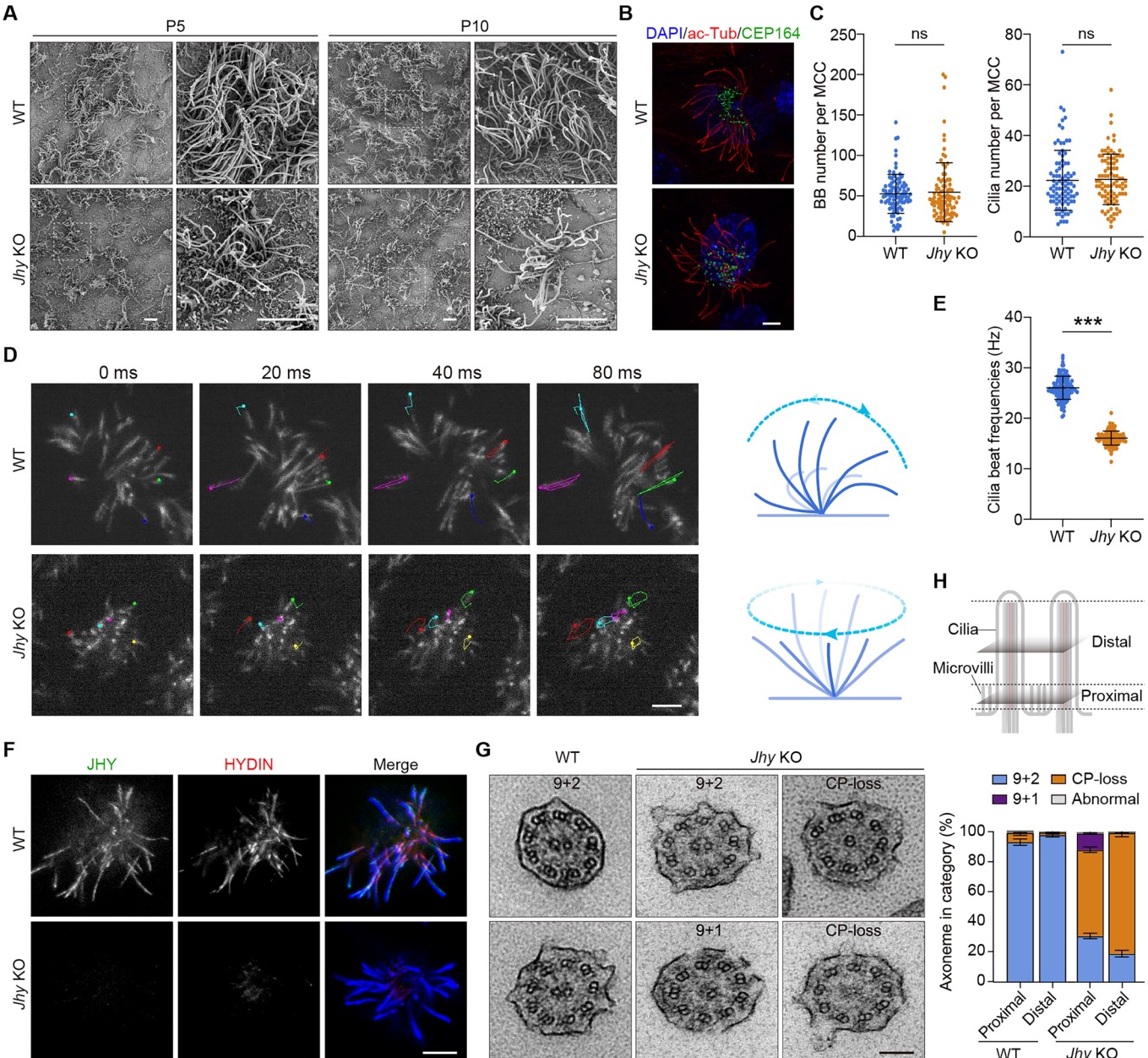

**Figure 5. JHY is required for the stabilization of CP-MTs.**

(A) Scanning electron microscopy analysis of ependyma isolated from WT and *Jhy* KO mice. Magnified images of the dashed boxed regions are shown on the right. Scale bar, 5 μm. (B, C) Immunofluorescence (B) and quantifications (C) of the number of basal bodies and cilia per multiciliated cell of WT and *Jhy* KO mEPCs. mEPCs were immunostained with the indicated antibodies and imaged with 3D-SIM. Ninety cells from three mice per genotype were scored using ImageJ. Data are presented as mean ± SD. Unpaired two-tailed *t* test was performed. ns, not significant (*P* = 0.6715 [BB]; *P* = 0.801 [Cilia]). Scale bar, 2 μm. (D, E) Representative frames (D) and quantifications (E) of ciliary beat frequency of WT and *Jhy* KO mEPCs. The trajectories of five cilia in each cell of the indicated genotypes are present. Diagrams illustrate the corresponding ciliary beat patterns. One hundred and twenty cells from three mice per genotype were scored using ImageJ. Data are presented as mean ± SD. Unpaired two-tailed *t* test was performed. *** (*P* < 0.0001). Scale bar, 5 μm. (F) Confocal images of WT and *Jhy* KO mEPCs immunostained with the indicated antibodies. Note that there is a complete loss of HYDIN in *Jhy* KO mEPCs. Scale bar, 5 μm. (G) Transmission electron microscopy analysis of WT and *Jhy* KO mEPCs. Scale bar, 100 nm. (H) Quantifications of indicated ciliary defects in the proximal and distal regions of WT and *Jhy* KO mEPCs. At least 180 cilia per genotype from three independent biological repeats were scored. A schematic diagram illustrates the definition of the proximal and distal regions of a motile cilium, defined by the presence or absence of microvilli around the axoneme. Data are from three independent biological repeats and presented as mean ± SD. Source data are available online for this figure.

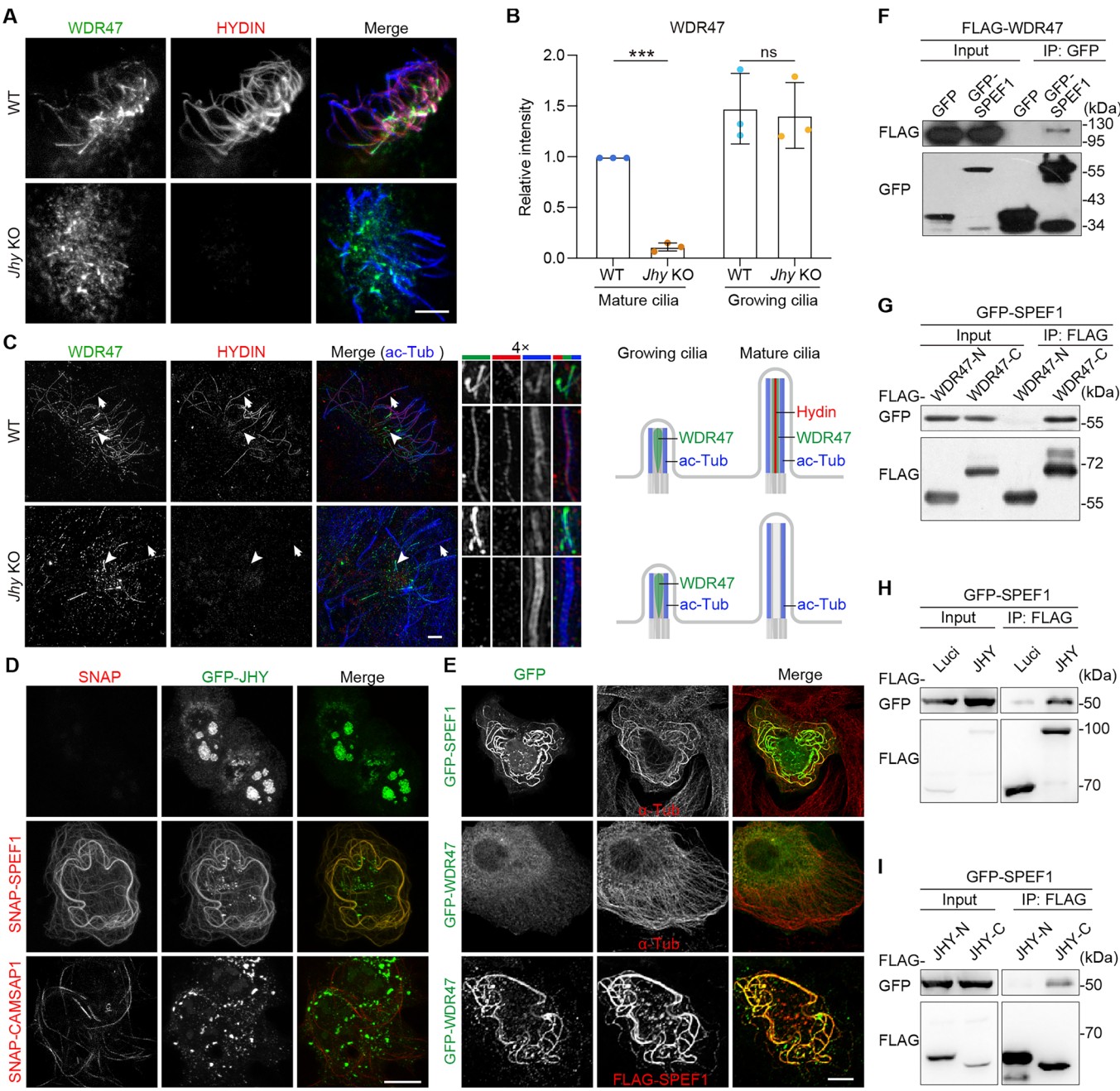

**Figure 6. JHY coordinates with WDR47 and SPEF1 to stabilize CP-MTs.**

(A) Confocal images of WT and *Jhy* KO mEPCs immunostained with the indicated antibodies. Scale bar, 5 μm. (B) Quantifications of the ciliary WDR47 intensity in the individual motile cilium of WT and *Jhy* KO mEPCs. At least 60 cilia per genotype from 3 independent biological repeats were scored. Data are presented as mean ± SD. Unpaired two-tailed *t* test was performed. *** (*P* < 0.0001); ns (*P* = 0.8202). (C) 3D-SIM images of WT and *Jhy* KO mEPCs immunostained with the indicated antibodies. Magnified images of mature long and growing short cilia, indicated by arrows and arrowheads, were shown on the right. Diagrams illustrate the spatial and temporal distributions of indicated proteins. Note that the ciliary tip accumulation of WDR47 was unaffected by JHY depletion in growing short cilia. Scale bar, 2 μm. (D, E) Confocal images of U2OS expressing the indicated proteins. Note that WDR47 and JHY can be recruited to the bundled microtubules induced by overexpression of SPEF1 (D). In contrast, no JHY signal exists in the bundled microtubules induced by CAMSAP1 overexpression (E). Scale bar, 10 μm. (F–I) Co-IP analysis in HEK293T cells expressing the indicated proteins exogenously. GFP-tagged proteins in (F) and Flag-tagged proteins in (G–I) were immunoprecipitated with anti-GFP and anti-Flag agarose beads, respectively. Blots were probed with the indicated antibodies. Source data are available online for this figure.

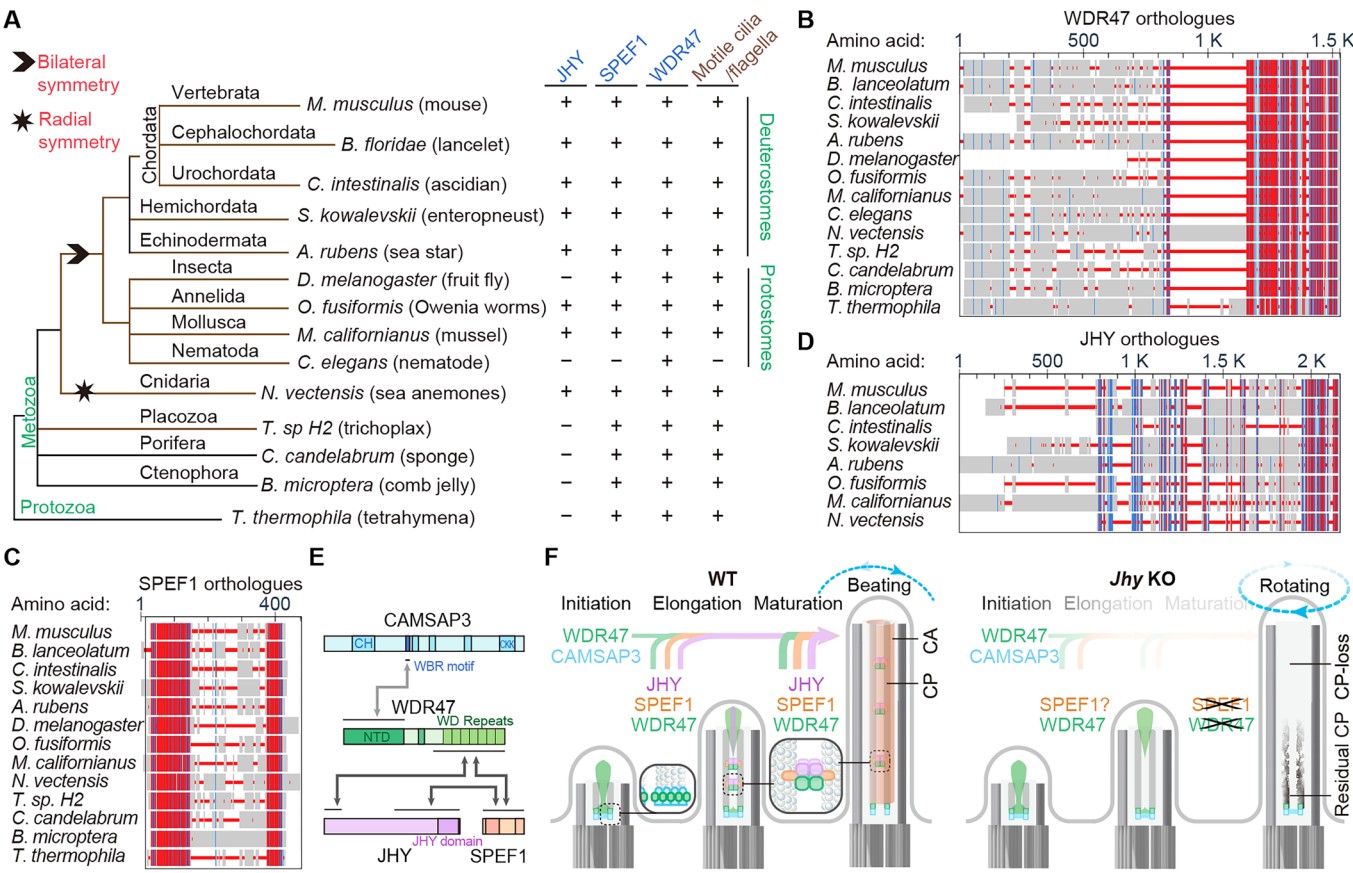

**Figure 7. Conservation of the WDR47, SPEF1, and JHY in evolution.**

(A) Conservation analysis of JHY, SPEF1, and WDR47 across species. (B–D) Similarities of WDR47 (B), SPEF1 (C), and JHY (D) orthologues were obtained using the constraint-based multiple alignment tool (COBALT) in NCBI. Highly conserved positions are highlighted in red, and less conserved positions in blue. Protein sequences of WDR47 orthologues used are: mouse (NP_852065), lancelet (CAH1272095), ascidian (XP_026695160), enteropneust (XP_006817534), sea star (XP_033643325), fruit fly (NP_611850), Owenia worms (CAH1785988), mussel (XP_052093602), nematode (NP_491864), sea anemones (XP_001635481), trichoplax (RDD44420), sponge (XP_062515027), comb jelly (XP_063684745), and tetrahymena (XP_001024878). Protein sequences of SPEF1 orthologues used are: mouse (Q99JL1), lancelet (XP_066277579), ascidian (XP_002127321), enteropneust (XP_006816413), sea star (XP_033633297), fruit fly (NP_573101), Owenia worms (CAH1783030), mussel (XP_052098468), sea anemones (XP_032230042), trichoplax (RDD38390), sponge (XP_062520407), comb jelly (XP_063690167), and tetrahymena (XP_001009857). Protein sequences of JHY orthologues used are: mouse (NP_001344288.1), lancelet (CAH1263516), ascidian (XP_018667279), enteropneust (XP_002731764), sea star (XP_033641474), Owenia worms (CAH1796106), mussel (XP_052061868), and sea anemones (XP_068685281). (E) Schematic diagrams illustrate the interactions of WDR47 with its interactors. (F) Schematic diagrams describe the findings in this study: (1) in nascent cilia, WDR47 is distributed to the central lumen to initiate CP-MT assembly; (2) as CP-MTs grow, SPEF1 binds to CP-MTs, where it recruits JHY and WDR47 to CP-MTs, and they cooperate to stabilize CP-MTs for proper ciliary motility; (3) in JHY-depleted cilium, although the initiation of CP-MT assembly by WDR47 is unaffected, *Jhy* deficiency causes the destabilization of CP-MTs and leads to severe CP-loss defects in the distal region of motile cilia, resulting in a rotational movement.

WDR47 and SPEF1 function as a dimer (Ren et al, 2022; Zheng et al, 2019), and that both WDR47 and JHY can be recruited to MT bundles by SPEF1, we propose that these three proteins form a multivalent complex that crosslinks and stabilizes CP-MTs. This stabilization supports CA projections assembly in growing cilia and ensures proper planar beating patterns in mature cilia (Fig. 7F).

Consistent with this model, CP-MTs are completely lost in *Wdr47*-deficient mice (Liu et al, 2021), while proximal CP-MTs partially persist in *Jhy*-deficient cilia (Figs. 5H and 7F). RNAi-mediated depletion of SPEF1 also severely disrupts CP-MTs (Zheng et al, 2019), although the phenotypes of a *Spef1* knockout mouse model have not been reported. Interestingly, in mouse mature cilia, WDR47 is enriched in the proximal CP-MT region, colocalizing with CAMSAPs (Liu et al, 2021), whereas JHY and SPEF1 are more evenly distributed along CP-MTs (Fig. 2) (Zheng et al, 2019),

suggesting that the WDR47-CAMSAP and WDR47-SPEF1-JHY complexes act independently. In *Xenopus* multiciliated cells, SPEF1 localizes to both the distal tips of mature cilia (Gray et al, 2009) (preprint: Hong and Lee 2025) and the apical cell membrane (Kim et al, 2018), indicating its multifunctional roles.

The conservation of WDR47 and SPEF1 (Fig. 7A–C) underscores the evolutionary importance of the WDR47-SPEF1 axis in CP-MT formation. Interestingly, insects appear to harbor two types of WDR47 orthologs. For instance, *D. melanogaster* (fruit fly) expresses a variant lacking the N-terminal CAMSAP-binding region (Fig. 7B), whereas *Bombyx mori* (silkworm) retains a full-length version (GenBank accession XP_004929817). Given that the N-terminal region of mouse WDR47 is critical for CAMSAP interaction (Chen et al, 2020; Ren et al, 2022), the *Drosophila* variant is predicted to lose the interaction with Patronin

(*Drosophila* CAMSAP) (Hendershott and Vale, 2014) and thus no longer participate in initial CP-MT assembly. Supporting this, CP-MTs in *Drosophila* sperm flagella are assembled from a single MT that is nucleated in the basal body of primary spermatocytes and stabilized by centriolar protein Bld10/CEP135 (Carvalho-Santos et al, 2012; Lattao et al, 2017), whereas silkworm flagellar CP-MTs start from two central MTs, following the canonical mechanism (Avidor-Reiss et al, 2020; Carvalho-Santos et al, 2012; Friedlander and Wahrman, 1971). Nonetheless, the C-terminal WD40 repeats-containing region, which interacts with SPEF1 and may recruit it to the central lumen (Figs. 6B and 7F) (Liu et al, 2021), is conserved in *Drosophila* (Fig. 7B), highlighting the critical role of the WDR47-SPEF1 axis even in species using divergent CP-MT formation mechanisms.

The absence of JHY from protozoa and early metazoa (Fig. 7A,D) suggests that, unlike WDR47 and SPEF1, JHY is not fundamentally required for CP-MT formation. Instead, it is evolved to enhance CP-MT stability in more complex animals (Fig. 7A), possibly by reinforcing the WDR47-SPEF1 complex (Fig. 7E,F). Many protists can switch beat waveforms of their cilia and flagella to detour in response to environmental stimuli (Marumo et al, 2021; Quarmby, 2009; Senatore et al, 2016). In contrast, motile cilia/flagella in vertebrates and many invertebrates beat in a fixed pattern, i.e., in either a "ciliary" waveform (a rapid effective stroke followed by a lazy recovery stroke) or a symmetric "flagellar" waveform (Senatore et al, 2016). Interestingly, Placozoans, Poriferans, and Ctenophores, organisms lacking JHY (Fig. 7A), retain the waveform flexibility (Leys and Degnan, 2001; Senatore et al, 2016; Smith et al, 2019; Tamm, 2014), whereas corals (Cnidarians), which contain JHY (Fig. 7A), exhibit a fixed waveform (Lewis and Price, 1976; Shapiro et al, 2014), suggesting that the emergence of JHY in evolution (Fig. 7A) may contribute to the transition from switchable to fixed waveforms. Consistently, JHY is lost in insects (Fig. 7A), in which only sperm flagella contain the CP-MTs (Keil, 2012; Lattao et al, 2017), and *Drosophila* flagella have been reported to display unusual abilities of switching waveforms (Lu, 2013). Future investigations are required to clarify the precise functions of JHY.

Additional CP-MT regulators may also exist. Due to the limitation of our screen, only WDR47-dependent regulators can be identified, which may be found among the additional proteins in the 42 candidates and the remaining hits in our proximity proteome (Fig. 1D,E). Indeed, a recent study has demonstrated the important role of CCDC13 in CP-MT formation (Wu et al, 2025). Thus, further investigation should be performed to assess the roles of other candidates, such as C2ORF81 and LRRC71 (Fig. 2).

# Methods

### Reagents and tools table

| Reagent/resource | Reference or source | Identifier or catalog number |
| --- | --- | --- |
| **Experimental models** | | |
| U2OS cells (*H. sapiens*) | ATCC | HTB-96 |
| HEK-293T cells (*H. sapiens*) | ATCC | CRL-11268 |
| HEK-293A cells (*H. sapiens*) | ThermoFisher | R70507 |

| Reagent/resource | Reference or source | Identifier or catalog number |
| --- | --- | --- |
| C57BL6/J (*M. musculus*) | GemPharmatech (Nanjing, China) | N000013 |
| *Jhy* KO mice | Shanghai Model Organisms Center | NM-KO-220140 |
| BL21-CodonPlus (DE3)-RIPL Competent Cells | Agilent | 230280 |
| **Recombinant DNA** | | |
| pLV-GFP-C1 | Zhao et al, 2013 | N/A |
| pAD-APEX2-Wdr47 | Liu et al, 2021 | N/A |
| pDONR221 | ThermoFisher | 12536017 |
| pEGX-4T-1 | NovoPro Bioscience | V010918 |
| pDEST-HisMBP | Addgene | 11085 |
| Gateway destination vectors | Addgene | 1000000211 |
| pLV-GFP-CCDC13 | This study | N/A |
| pLV-GFP-CCDC108 | This study | N/A |
| pLV-GFP-C2ORF81 | This study | N/A |
| pLV-GFP-ENO4 | This study | N/A |
| pLV-GFP-LRGUK | This study | N/A |
| pLV-GFP-LRRC71 | This study | N/A |
| pLV-GFP-MYCBPAP | This study | N/A |
| pDONR221-JHY | This study | N/A |
| pDONR221-JHYN | This study | N/A |
| pDONR221-JHYC | This study | N/A |
| pDONR221-JHYN1 | This study | N/A |
| pDONR221-JHYM | This study | N/A |
| pDONR221-WDR47 | This study | N/A |
| pDONR221-WDR47N | This study | N/A |
| pDONR221-WDR47C | This study | N/A |
| pDONR221-WDR47-M | This study | N/A |
| pDONR221-WDR47C1 | This study | N/A |
| pDONR221-SPEF1 | This study | N/A |
| pEGX-4T-1-JHYN | This study | N/A |
| **Antibodies** | | |
| Rabbit anti-Kif9 antibody | Fang et al, 2024 | N/A |
| Rabbit anti-WDR47 antibody | Liu et al, 2021 | N/A |
| Rabbit anti-Rsph4a antibody | Zhu et al, 2019 | N/A |
| Guiea Pig anti-Hydin antibody | This study | N/A |
| Rabbit anti-JHY antibody | This study | N/A |
| Rat anti-HYDIN antibody | This study | N/A |
| Guinea pig anti-CEP164 antibody | This study | N/A |
| Mouse anti-acetylated tubulin antibody | Sigma-Aldrich | T6793 |
| Rat anti-GFP antibody | BioLegend | 338002 |
| Mouse anti-GFP antibody | Roche | 11814460001 |
| Rabbit anti-FLAG antibody | Abclonal | AE092 |
| Rabbit anti-GFP antibody | Abclonal | AE011 |

  

| Reagent/resource | Reference or source | Identifier or catalog number |
|---|---|---|
| Mouse anti-acetylated tubulin antibody | Proteintech | 66200-1-Ig |
| HRP-conjugated His-Tag Antibody | Proteintech | HRP-66005 |
| HRP-conjugated GST-Tag Antibody | Proteintech | HRP-66001 |
| Goat anti-Rabbit IgG (H + L) | Abberior | STORANGE-1002 |
| Goat anti-Mouse IgG (H + L) | Abberior | STORANGE-1001 |
| Goat anti-Rabbit IgG (H + L) | Abberior | STRED-1002 |
| Goat anti-Mouse IgG (H + L) | Abberior | STRED-1001 |
| Alexa Fluor Plus 488-conjugated Rabbit anti-GFP antibody | ThermoFisher | A21311 |
| Alexa Fluor Plus 488-conjugated Chicken anti-GFP antibody | ThermoFisher | A10262 |
| Pacific Blue-conjugated Goat anti-Mouse IgG (H + L) | ThermoFisher | P31582 |
| Alexa Fluor 555-conjugated Donkey anti-Mouse IgG (H + L) | ThermoFisher | A31570 |
| Alexa Fluor Plus 488-conjugated Donkey anti-Rabbit IgG (H + L) | ThermoFisher | A32790 |
| Alexa Fluor 488-conjugated Goat anti-Chicken IgY (H + L) | ThermoFisher | A11039 |
| HRP-conjugated Goat anti-Rabbit IgG (H + L) | ThermoFisher | 31460 |
| Cy3-conjugated Donkey anti-Guinea pig IgY (H + L) | Jackson ImmunoResearch | 706-165-148 |
| Alexa Fluor 647-conjugated Donkey anti-Guinea pig IgY (H + L) | Jackson ImmunoResearch | 706-605-148 |
| DyLight 405-conjugated Donkey anti-Mouse IgG (H + L) | Jackson ImmunoResearch | 715-475-151 |
| Alexa Fluor 555-conjugated Donkey anti-Rat IgG (H + L) | Jackson ImmunoResearch | 712-565-153 |
| Alexa Fluor 488-conjugated Donkey anti-Guinea pig IgG (H + L) | Jackson ImmunoResearch | 706-545-148 |
| **Oligonucleotides and other sequence-based reagents** | | |
| PCR primers | This study | Table EV1 |
| **Chemicals, enzymes, and other reagents** | | |
| Polyethylenimine (PEI) | Polysciences | 23966 |
| Neutral resin | Solarbio | G8590 |
| Papain | Worthington | LS003126 |
| Biotin-phenol | Iris Biotech | LS-3500.1000 |
| Ni-NTA agarose beads | Qiagen | 30210 |
| GFP-Nanoab-Agarose beads | Lablead | GNA-50-1000 |
| glycogen PAS staining kit | KeyGen Biotech | KGE1103-400 |
| ClonExpress II One Step Cloning Kit | Vazyme | C112 |
| 2 × Taq Plus Master Mix II | Vazyme | P213 |
| Gateway™ LR Clonase™ II Enzyme mix | ThermoFisher | 11791100 |
| HRP-conjugated Streptavidin | ThermoFisher | S911 |
| DMEM high glucose | ThermoFisher | C11995500BT |

| Reagent/resource | Reference or source | Identifier or catalog number |
|---|---|---|
| Fetal bovine serum (FBS) | ThermoFisher | A5256701 |
| Lipofectamine 2000 | ThermoFisher | 11668019 |
| Streptavidin-conjugated agarose beads | Sigma-Aldrich | S1638 |
| Glutathione sepharose beads | Sigma-Aldrich | GE17075601 |
| Complete protease inhibitors | Sigma-Aldrich | 539134 |
| Fibronectin | Sigma-Aldrich | FC010 |
| **Software** | | |
| GraphPad Prism | https://www.graphpad.com | |
| ImageJ | https://imagej.nih.gov/ij/index.html | |
| **Other** | | |
| Leica TCS SP8 system | Leica | |
| Delta Vision OMX SR imaging system | GE Healthcare | |
| Abberior STEDYCON STED microscope | Abberior | |
| Olympus SpinSR10 microscope | Olympus | |
| Olympus SZX16 stereo microscope | Olympus | |
| RM 2255 microtome | Leica | |
| Leica VT 1000S vibratome | Leica | |
| T7800 transmission electron microscop | Hitachi Asia Ltd | |
| TM3030 scanning electron microscope | Hitachi Asia Ltd | |

## Plasmids

Full-length mouse *Jhy* (NM_001357359), *Wdr47* (NM_181400), *Spef1* (NM_027641), *Ccdc13* (NM_028384), *Ccdc108* (NM_001039495), *C2orf81* (NM_027948), *Eno4* (NM_178689), *Lrguk* (NM_028886), *Lrrc71* (NM_028971), and *Mycbpap* (NM_170671) were amplified from a mouse cDNA library by polymerase chain reaction (PCR) and subcloned into the lentiviral expression vector pLV-EGFP-C1 using the ClonExpress II One Step Cloning Kit. The expression constructs were obtained from LR recombination reactions between entry clones and desired Gateway destination vectors. To generate tag-fused constructs, full-length and specified fragments were PCR-amplified and subcloned into the donor vector pDONR221, followed by LR recombination reactions between entry clones and the desired Gateway destination vectors. To generate bacterial expression constructs, the indicated *Wdr47* and *Jhy* fragments were subcloned into pGEX-4T-1 and pDEST-HisMBP. To generate the adenoviral plasmid expressing APEX2-WDR47, full-length *Wdr47* was subcloned into pLV-APEX2, and the *APEX2-Wdr47* fused gene was used to construct the adenoviral plasmid as previously described (Zhao et al, 2021). All constructs were verified via Sanger sequencing analysis. All the primers used are listed in Table EV1.

## Cell culture and transfection

Human HEK293T, HEK293A, and U2OS cells were cultured in Dulbecco's modified Eagle's medium (DMEM) supplemented with 10% fetal bovine serum (FBS), 1% penicillin/streptomycin, and 2 mM L-alanyl-l-glutamine. Cell lines have not been authenticated since their purchase, and all cell lines were routinely tested for mycoplasma contamination. mEPCs were cultured as previously described (Zhao et al, 2019a). In brief, the telencephala were isolated from neonates (P0-P3) and digested with papain to obtain mEPC precursors. After removing neural cells by mechanical shake-off, precursor cells were cultured to ~80% confluency and then seeded onto the fibronectin-coated cover glass for immunofluorescence staining or placed into a fibronectin-coated glass-bottom dish for live imaging. Upon 100% confluency, cells were serum-starved to initiate differentiation. HEK293T cells were transfected with the indicated plasmids and 1 mg/ml polyethylenimine (PEI) at a ratio of 2:3 and harvested 48 h after transfection. U2OS cells were transfected with the indicated plasmids using Lipofectamine 2000.

## Virus preparation and infection

Lentiviral particles and adenovirus were produced as described previously (Zhao et al, 2013). In brief, a lentiviral plasmid, pCMV-D8.9, and pCMV-VSVG were transfected into HEK293T cells at the ratio of 5:3:2 using PEI for 48 h. The culture medium containing lentiviral particles was collected and added to the mEPC culture medium at a 1:20 dilution. To produce adenovirus, the recombinant adenoviral plasmid containing the indicated gene was linearized and transfected into HEK293A cells using Lipofectamine 2000. The first-generation adenovirus was harvested when HEK293A cells displayed obvious morphological changes (cytopathic effect). The adenovirus was amplified by infecting fresh HEK293A cells and tested for protein expression in HEK293T cells by immunoblotting. Adenovirus was diluted into the culture medium for the infection of mEPCs.

## Proximity labeling assay

APEX2-mediated biotin-phenol labeling and enrichment were performed as described previously (Chen et al, 2025). mEPCs were infected with adenovirus expressing APEX2-WDR47 at day -2 and serum-starved to induce ciliogenesis for ten days. To initiate proximity labeling, cells were incubated with 500 μM biotin-phenol at 37 °C for 30 min, followed by the addition of 1 mM $H_2O_2$ for 1 min. The labeling reaction was quenched by adding PBS containing 10 mM sodium ascorbate, 10 mM sodium azide, and 5 mM Trolox. For immunostaining, biotinylated proteins were labeled with Alexa Fluor 546-conjugated streptavidin. To enrich biotinylated proteins, mEPCs were lysed with RIPA buffer supplemented with 1 mM PMSF, 1 mM dithiothreitol (DTT), and complete protease inhibitors, and the supernatants were incubated with streptavidin-conjugated agarose beads for 4 h at 4 °C. The biotinylated proteins were incubated with 2× protein loading buffer supplemented with 20 mM DTT and 2 mM biotin and boiled for 5 min. The eluted samples were subjected to mass spectrometry analysis. The functional enrichment analysis was conducted using Metascape, and the proteomic analysis was visualized using the ggplot2 and ggrepel packages in R.

## Immunoprecipitation

Coimmunoprecipitation was conducted as described previously (Song et al, 2025; Zhao et al, 2020). HEK293T cells exogenously expressing indicated proteins were lysed in high-salt lysis buffer (1% NP-40, 500 mM NaCl, 50 mM HEPES [PH 7.8], 5 mM EDTA) containing 50 mM NaF, 1 mM $Na_3VO_4$, 3 mM DTT, 1 mM PMSF, and complete protease inhibitors. Cell contents were further released using a tissue homogenizer and cleared by centrifugation at $14,000 \times g$ for 10 min at 4 °C. The supernatants were incubated with GFP-Nanoab-Agarose beads at 4 °C for 4 h with gentle rotation. Beads were washed thrice in lysis buffer, and proteins were eluted into a sample buffer (50 mM Tris, 2% sodium dodecyl sulfate [SDS], 10% glycerol, 5% β-mercaptoethanol, PH 6.8) and analyzed by SDS polyacrylamide gel electrophoresis (SDS-PAGE).

## GST pulldown assay

His-MBP-JHYN1 and GST-WDR47C1 fragment proteins were expressed using BL21-Codon Plus (DE3)-RIPL cells. His-tagged proteins were purified using Ni-NTA agarose beads. Bacterial cells expressing GST or GST-WDR47C1 were lysed with PBS containing 1% Triton X-100 and cleared by centrifugation at $14,000 \times g$ for 10 min at 4 °C. The supernatants were incubated with glutathione sepharose beads at 4 °C for 4 h with gentle rotation. After being washed thrice with high-salt lysis buffer, beads with GST or GST-WDR47C1 protein were incubated with 1 ml high-salt lysis buffer containing purified His-MBP-JHYN1 for 4 h at 4 °C. Enriched proteins were eluted in a sample buffer and analyzed by SDS-PAGE.

## Fluorescence microscopy

Immunostaining of mEPCs on cover glasses was conducted as described previously (Zhao et al, 2019a; Zhu et al, 2019). Briefly, cells were fixed with freshly prepared 4% PFA in PBS for 15 min at room temperature and permeabilized with 0.5% Triton X-100 in PBS for 15 min. Samples were then blocked with 4% bovine serum albumin (BSA) in tris-buffered saline with 0.1% Tween 20 (TBST) for 1 h. Primary and secondary antibodies diluted in the blocking buffer were incubated with cells at 4 °C overnight and at room temperature for 1 h, respectively. All the antibodies used are listed in Table EV2.

Confocal images were captured using a Leica TCS SP8 system with a ×63/1.40 oil-immersion objective, and Z-stack images were obtained with maximum intensity projections. 3D-SIM images were obtained using a Delta Vision OMX SR imaging system (GE Healthcare) with serial Z-stack sectioning at 125-nm intervals. 3D-SIM images were processed with SoftWoRx. STED imaging was performed on an Abberior STEDYCON STED microscope with a ×100/1.5NA oil-immersion objective (Evident Scientific). STED images were deconvolved using Huygens deconvolution software (Huygens Essentials). Ciliary motilities were recorded at 140 frames per second (fps) using an Olympus IX71 microscope equipped with an Andor Neo sCMOS camera and a ×63/1.40 oil-immersion objective. Images were processed with ImageJ.

## Animals

Mice experiments were performed in accordance with the ethical guidelines approved by the Institutional Animal Care and Use

Committee of Shandong Normal University (AEECSDNU2024082). Mice were housed in groups (2–5 per cage) and maintained in a temperature-controlled facility featuring a 12:12 light/dark cycle and a pathogen-free environment. The *Jhy* KO founder mice were purchased from Shanghai Model Organisms Center, Inc., which were generated using the CRISPR-Cas9 system with dual sgRNAs to delete the genomic sequence containing the exon 3 of the mouse *Jhy* gene (Transcript ID: ENSMUSG00000032023). *Jhy* founder mice were intercrossed to obtain KO mice. Mice were genotyped using 2× Taq Plus Master Mix II. Quantitative real-time PCR was conducted as previously described (Lu et al, 2025; Zi et al, 2024). *Gapdh* was utilized for normalization. Primers used are listed in Table EV1.

## Tissue preparation and histological analysis

Mice at P10 were euthanized by decapitation. Brains were dissected and sectioned into 250-μm-thick sagittal slices using a Leica VT 1000S vibratome. Images were captured with an Olympus SZX16 Stereo Microscope. For histological analysis of nasal cavities, mice at P10 were euthanized by decapitation. Trimmed heads were skinned and fixed with 4% PFA in PBS overnight, followed by decalcification in 30% hydrochloric acid solution with 2% acetic acid and 3% PFA overnight. After decalcification, samples were paraffin-embedded and sectioned into slices of 5 μm thickness with an RM 2255 microtome (Leica). The periodic acid-Schiff (PAS) staining of dewaxed sections was performed using the glycogen PAS staining kit. Briefly, sections were immersed in the hematoxylin staining solution for 30 s and then placed in 1% hydrochloric acid ethanol solution for 5 s. After sequential dehydration, the sections were immersed in xylene twice for transparency and mounted with the neutral resin.

## Electron microscopy

For scanning electron microscopy, ependyma isolated from WT and *Jhy* KO mice were fixed with 2.5% glutaraldehyde (GA) and 4% PFA overnight and post-fixed in 1% $OsO_4$ for 1 h. Samples were then dehydrated with gradient ethanol (50%, 70%, 80%, 90%, 100%, 100%, and 100%) for 10 min each. Sequentially, samples were dried by critical point drying, gold-coated by the sputtering technique, and examined with a scanning electron microscope at an accelerating voltage of 15 kV. For transmission electron microscopy, mEPCs were fixed in 2.5% GA and 4% PFA for 1 h at room temperature, followed by fixation at 4 °C overnight. Samples were washed with PBS, post-fixed with 1% OsO4 for 1 h at 4 °C, dehydrated through a graded ethanol series, and embedded in Epon 812 resin. Samples were sectioned at a thickness of 70 nm and stained with 2% uranyl acetate for 10 min and 1% lead citrate for 5 min. Images were captured at 80 KV using an HT7800 transmission electron microscope (Hitachi Asia Ltd).

## Statistical analysis

All experiments were biologically repeated at least three times. In total, 3–6 mice per genotype were randomly used for each experiment. No blinding was done for the analyses. One representative picture from three to six mice for each genotype was presented for immunostaining. Numbers of basal bodies and cilia were measured using 3D-SIM images. The ciliary beat frequency and relative intensity of indicated proteins in immunostaining images were measured using ImageJ. The quantitative results were presented as the mean ± standard deviation (SD) unless specified in the figure legend. Statistical analyses were conducted using GraphPad Prism. Unpaired two-tailed Student's *t* tests were used for comparison between the two groups. Differences were considered significant when $P < 0.05$.

## Data availability

No data were deposited in external repositories for this study.

The source data of this paper are collected in the following database record: biostudies:S-SCDT-10_1038-S44319-025-00671-7.

## Peer review information

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

## Acknowledgements

We thank Heng Guo (Electron Microscopy Core) and Ying Li (Light Microscopy Core) at Shandong Normal University for their assistance in imaging. We thank the institutional core facility of cell biology at the Shanghai Institute of Biochemistry and Cell Biology for instrumental and technical support and the National Facility for Protein Science Shanghai for providing mass spectrometry. We thank Mr. Ji Tang (OptoFeM Technology) for assisting with STED imaging. This work was supported by the National Natural Science Foundation of China (32230027, 32270807, and 32500663), the Shandong Natural Science Foundation (ZR2024QC038), and the Taishan Scholar Foundation of Shandong Province (tsqn202211109).

## Author contributions

**Qingxia Chen**: Validation; Investigation; Writing—original draft. **Shuxiang Ma**: Validation; Investigation. **Hao Liu**: Investigation. **Juyuan Liu**: Investigation. **Qingchao Li**: Investigation; Visualization. **Qian Lyu**: Validation. **Hanxiao Yin**: Formal analysis. **Junkui Zhao**: Validation. **Shanshan Nai**: Funding acquisition; Validation. **Ting Song**: Funding acquisition; Investigation. **Hongbin Liu**: Resources. **Jun Zhou**: Resources. **Xiumin Yan**: Supervision. **Xueliang Zhu**: Conceptualization; Supervision; Funding acquisition; Writing—review and editing. **Huijie Zhao**: Conceptualization; Supervision; Funding acquisition; Writing—original draft; Project administration; Writing—review and editing.

Source data underlying figure panels in this paper may have individual authorship assigned. Where available, figure panel/source data authorship is listed in the following database record: biostudies:S-SCDT-10_1038-S44319-025-00671-7.

## Disclosure and competing interests statement

The authors declare no competing interests.

# Expanded View Figures

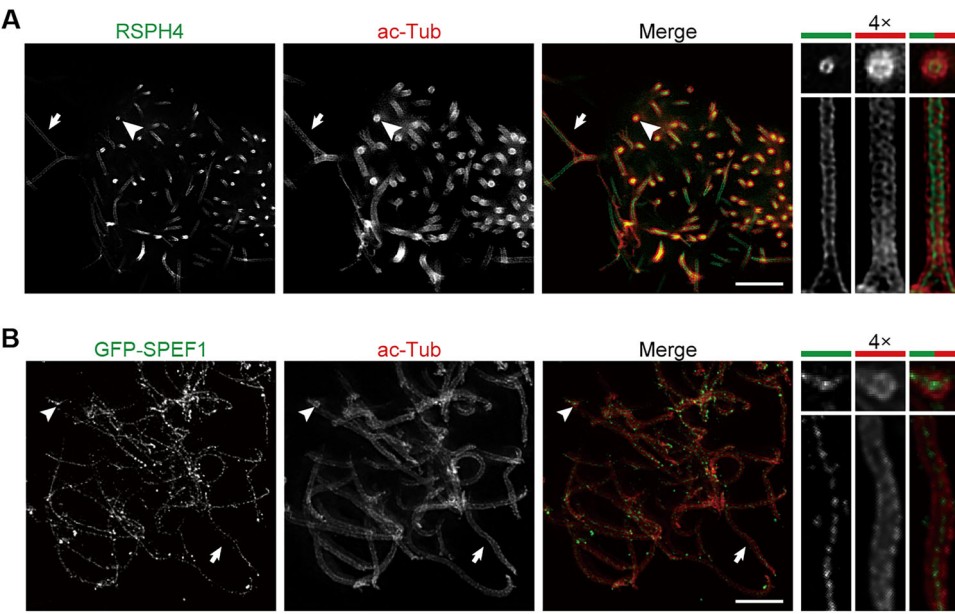

**Figure EV1. Discovery of new central lumen proteins with super-resolution microscopy.**

(A) STED images of mEPCs immunostained with the indicated antibodies. Cells were serum-starved to induce multiciliogenesis and labeled with antibodies against acetylated α-tubulin (ac-Tub) and RSPH4. Magnified images on the right show the longitudinal and transverse views, indicated by arrows and arrowheads, respectively. Note that RSPH4 displays a ring-like distribution in the central lumen. Scale bar, 2 μm. (B) STED images of mEPCs exogenously expressing GFP-SPEF1. Cells were serum-starved to induce multiciliogenesis and labeled with an acetylated α-tubulin (ac-Tub) antibody. Magnified images on the right show the longitudinal and transverse views indicated by arrows and arrowheads. Scale bar, 2 μm. Source data are available online for this figure.

