## [Peer Review File · EMBO Reports]

JHY enables the transition from switchable to fixed ciliary waveforms in metazoan evolution

Qingxia Chen, Shuxiang Ma, Hao Liu, Juyuan Liu, Qingchao Li, Qian Lyu, Hanxiao Yin, Junkui Zhao, Shanshan Nai, Ting Song, HongBin Liu, Jun Zhou, Xiumin Yan, Xueliang Zhu, and Huijie Zhao

Corresponding author(s): Huijie Zhao (huijiezhao@sdu.edu.cn) , Xueliang Zhu (xlzhu@sibcb.ac.cn)

Review Timeline:

Submission Date:	19th Jun 25
Editorial Decision:	18th Aug 25
Revision Received:	2nd Oct 25
Editorial Decision:	10th Nov 25
Revision Received:	11th Nov 25
Accepted:	26th Nov 25

Editor: Deniz Senyilmaz Tiebe / Kurt Weir

Transaction Report:

Dear Prof. Zhao,

Thank you for transferring your manuscript to EMBO Reports, which was now seen by two referees, whose reports are copied below. I apologize for this unusual delay in getting back to you. It took longer than anticipated to receive the full set of referee reports given this busy time of the year.

Referees express interest in the proposed role of JHY in the transition from switchable to fixed ciliary waveforms in metazoa. However, they also raise some concerns that need to be addressed to consider publication here.

I find the reports informed and constructive, and believe that addressing the concerns raised will significantly strengthen the manuscript. As the reports are below, and I think all points need to be addressed, I will not detail them here. Please contact me if you have questions or comments regarding the revision for further discussion (also by video chat).

Should you be able to address all referee concerns, we would like to invite you to revise your manuscript with the understanding that the referee concerns (as in their reports) must be fully addressed and their suggestions taken on board. Please address all referee concerns in a complete point-by-point response. Acceptance of the manuscript will depend on a positive outcome of a second round of review. It is EMBO reports policy to allow a single round of major experimental revision only and acceptance or rejection of the manuscript will therefore depend on the completeness of your responses included in the next, final version of the manuscript.

We realize that it is difficult to revise to a specific deadline. In the interest of protecting the conceptual advance provided by the work, we recommend a revision within 3 months. Please discuss the revision progress ahead of this time with me if you require more time to complete the revisions, or if you have questions or comments regarding the revision (also by video chat).

1. A data availability section providing access to data deposited in public databases is missing (where applicable).
2. Your manuscript contains statistics and error bars based on $n=2$. Please use scatter plots in these cases.

You can submit the revision either as a Scientific Report or as a Research Article. For Scientific Reports, the revised manuscript can contain up to 5 main figures and 5 Expanded View figures, and it should not exceed 27000 characters. If the revision leads to a manuscript with more than 5 main figures it will be published as a Research Article. In this case the Results and Discussion section should be separate. If a Scientific Report is submitted, these sections have to be combined. This will help to shorten the manuscript text by eliminating some redundancy that is inevitable when discussing the same experiments twice. In either case, all materials and methods should be included in the main manuscript file.

3) We replaced Supplementary Information with Expanded View (EV) Figures and Tables that are collapsible/expandable online. A maximum of 5 EV Figures can be typeset. EV Figures should be cited as 'Figure EV1, Figure EV2' etc... in the text and their respective legends should be included in the main text after the legends of regular figures.

4) a .docx formatted letter INCLUDING the reviewers' reports and your detailed point-by-point responses to their comments. As part of the EMBO publication's Transparent Editorial Process, EMBO reports publishes online a Review Process File (RPF) to

accompany accepted manuscripts. This File will be published in conjunction with your paper and will include the referee reports, your point-by-point response and all pertinent correspondence relating to the manuscript.

<https://www.embopress.org/page/journal/14693178/authorguide#transparentprocess>

5) a complete author checklist, which you can download from our author guidelines

<https://www.embopress.org/page/journal/14693178/authorguide>. Please insert information in the checklist that is also reflected in the manuscript. The completed author checklist will also be part of the RPF.

6) Please note that all corresponding authors are required to supply an ORCID ID for their name upon submission of a revised manuscript (<<https://orcid.org/>>). Please find instructions on how to link your ORCID ID to your account in our manuscript tracking system in our Author guidelines

<<https://www.embopress.org/page/journal/14693178/authorguide#authorshipguidelines>>

7) Before submitting your revision, primary datasets produced in this study need to be deposited in an appropriate public database (see <https://www.embopress.org/page/journal/14693178/authorguide#datadeposition>). Please remember to provide a reviewer password if the datasets are not yet public. The accession numbers and database should be listed in a formal "Data Availability" section placed after Materials & Method (see also

<https://www.embopress.org/page/journal/14693178/authorguide#datadeposition>). Please note that the Data Availability Section is restricted to new primary data that are part of this study. * Note - All links should resolve to a page where the data can be accessed. *

Additional information on source data and instruction on how to label the files are available:

<https://www.embopress.org/page/journal/14693178/authorguide#sourcedata>

9) Our journal encourages inclusion of *data citations in the reference list* to directly cite datasets that were re-used and obtained from public databases. Data citations in the article text are distinct from normal bibliographical citations and should directly link to the database records from which the data can be accessed. In the main text, data citations are formatted as follows: "Data ref: Smith et al, 2001" or "Data ref: NCBI Sequence Read Archive PRJNA342805, 2017". In the Reference list, data citations must be labeled with "[DATASET]". A data reference must provide the database name, accession number/identifiers and a resolvable link to the landing page from which the data can be accessed at the end of the reference. Further instructions are available at <http://www.embopress.org/page/journal/14693178/authorguide#referencesformat>

10) Regarding data quantification (see Figure Legends:

<https://www.embopress.org/page/journal/14693178/authorguide#figureformat>)

11) The journal requires a statement specifying whether or not authors have competing interests (defined as all potential or actual interests that could be perceived to influence the presentation or interpretation of an article). In case of competing interests, this must be specified in your disclosure statement. Further information: <https://www.embopress.org/competing->

interests

12) Please also note our reference format:

13) All Materials and Methods need to be described in the main text using our 'Structured Methods' format, which is required for all research articles. According to this format, the Methods section includes a Reagents and Tools Table (listing key reagents, experimental models, software and relevant equipment and including their sources and relevant identifiers) followed by a Methods and Protocols section describing the methods using a step-by-step protocol format. The aim is to facilitate adoption of the methodologies across labs. More information on how to adhere to this format as well as a downloadable template (.docx) for the Reagents and Tools Table can be found in our author guidelines:

I look forward to seeing a revised version of your manuscript when it is ready. Please let me know if you have questions or comments regarding the revision.

Kind regards,

Deniz Senyilmaz Tiebe

Deniz Senyilmaz Tiebe, PhD
Senior Scientific Editor
EMBO Reports

Referee #1:

This manuscript presents a comprehensive study on the functional significance of the Juvenile hydrocephalus (JHY) protein. While previous research identified the *jhy* gene as a key factor leading to rapidly progressive hydrocephalus and revealed structural defects in cilia due to its absence, the precise structural and biological roles of the JHY protein itself have remained unclear.

The authors convincingly demonstrate that JHY is essential for the formation of the central apparatus (CA) in functional motile cilia of multiciliated mouse endothelial progenitor cells (mEPCs). The conclusions are supported by a robust and multi-faceted experimental approach, including:

1. Proximity proteomics using WDR47, a central regulator of central pair microtubule (CP-MT) formation and maintenance, as bait.
2. Subcellular localization studies-using both high-resolution fluorescence microscopy and antibody-based detection-to track JHY and CP-MT-associated proteins.
3. Co-immunoprecipitation assays employing GFP-tagged, FLAG-tagged, and deletion mutants to validate the interaction between JHY and WDR47.
4. Gene expression analysis of *jhy* across different tissues via qPCR.
5. Phenotypic characterization of *jhy* knockout mice, including quantification of basal bodies and cilia per cell, assessment of ciliary beat frequency, and immunohistochemical analysis of HYDIN, WDR47, and SPEF1 in mEPCs.
6. Phylogenetic analysis of *jhy* orthologs to explore evolutionary conservation.

I am highly impressed by the quality and clarity of the experimental work. The data are exceptionally clean and the logical flow of the experimental design and result interpretation is outstanding-reminiscent of textbook-level clarity, which is rarely encountered in peer-reviewed articles.

The manuscript is well written, the English is easy to follow, and the figures are carefully constructed with informative legends.

One potential next step for the authors would be to perform a high-resolution structural analysis of the JHY protein, such as through cryo-electron microscopy or X-ray crystallography. Such structural insights could further elucidate the molecular mechanisms by which JHY contributes to central apparatus formation and function. However, I recognize that this would be a

substantial undertaking beyond the scope of the current study and is not necessary for publication in EMBO Reports. The current dataset is sufficiently complete and compelling to warrant publication as is.

One very minor point: while it is evident that jhy refers to Juvenile hydrocephalus, the authors should define this abbreviation explicitly at its first appearance in the main text.

Aside from this ignorable minor correction, I strongly recommend publication of this manuscript in EMBO Reports without further revision.

Referee #2:

EMBOR-2025-62146-T

"JHY enables the transition from switchable to fixed ciliary waveforms in metazoan evolution"

Summary

• 1. Does this manuscript report a single key finding?

YES

JHY interacts with SPEF1 and WDR47 and plays a role in central pair microtubule stabilization in mouse ependymal cilia.

• 2. Is the reported work of significance (YES), or does it describe a confirmatory finding or one that has already been documented using other methods or in other organisms etc?

YES

The significance of JHY in CP-MT assembly/stability was not earlier investigated.

1. 3. Is it of general interest to the molecular biology community?

YES

The mechanism(s) enabling non-centrosomal CP-MT formation/stability, indispensable for proper functioning of motile cilia, is poorly understood despite several recently published papers.

• 4. Is the single major finding robustly documented using independent lines of experimental evidence (YES), or is it really just a preliminary report requiring significant further data to become convincing, and thus more suited to a longer-format article (NO)?

YES

While searching for proteins involved in CP-MT assembly, the authors expressed WDR47, a key CP assembly factor, in fusion with APEX2 and performed a proximity assay. The comparison of the putative Wdr47 interactome encompassing 2448 proteins with Wdr47 knockout ciliomes led to the identification of 43 candidate proteins both enriched in the WDR47 interactome and significantly reduced in KO. After localization studies of some of the identified CP components or proteins of unknown intraciliary position, the authors focused their interest on JHY, a protein earlier shown to be linked to the CP-MT formation.

Using microscopic and biochemical approaches, the authors showed that JHY directly interacts with WDR47 and SPEF1, a protein known to bind to MT seam to stabilize MTs, and that these proteins co-localize in mouse ependymal cilia in a cilia-length-dependent manner. The JHY knockout mice exhibited PCD-related phenotype, including hydrocephalus. Cilia assembled by Jhy KO mEPCs displayed rotatory movement and a CP-loss, especially in the distal part of the cilia. Based on collected data, the authors concluded that SPEF1 recruits JHY and WDR47 to central MTs and that JHY plays a role in CP-MT stabilization. Identification of JHY as a protein that directly interacts with WDR47 and SPEF1 is a new and interesting finding, significantly contributing to our understanding of the molecular mechanism of CP-MT assembly and stability. The provided biochemical and microscopic data are convincing and surely will be of interest to all who study motile cilia or cilia-related diseases.

However, there are also some weak points in this manuscript. The manuscript can be shortened and perhaps the order of the chapters reorganized. The last chapter describing phylogenetic analyses should be shorter, more concise, and perhaps some parts should be combined with the relevant discussion section.

The authors suggested that obtained proximity proteomics data can be used to identify components of CA projection. However, the putative WDR47 interactome containing 2448 proteins (only 14% are ciliary proteins) is not sufficient to select such candidates. Even among ciliary proteins, one can find numerous doublet proteins. If the JHY was not earlier shown to be related to CP formation, would the authors select JHY for further analyses based exclusively on WDR47 interactome and WDR47 KO data?

Major comments:

Controls for the APEX2 experiment (WT cilia, cilia from cells expressing only APEX2) showing a background biotinylation are missing. It is surprising that only 14% of the proteins identified in this experiment were annotated as ciliary proteins. Perhaps this is also due to the identification of some non-specifically biotinylated proteins. Did the authors try to optimize assay conditions? Without additional data, the presented interactome is rather insufficient to identify remaining CA projection components as the authors suggest at the end of the Discussion.

While in the Dataset EV1, the identified proteins are ordered according to the decreasing number of peptides, the reason for ordering proteins in Dataset EV4 as they are ordered is unclear (proteins are neither in alphabetical order (protein names or accession number) nor according to the number of peptides).

I understand that it is not easy, but can the authors next include graphs showing changes in the intensity of IF signals along the

cilia length to support their model of protein localization during cilia assembly and elongation?

Missing REFs:

- page 7: Studies in a model organism, *Tetrahymena*, showed that ADGB is a subunit of C1b projection (Joachimiak et al., 2021); those findings were supported by Qu et al., showing direct interactions between ADGB and two C1b proteins, CFAP69 and SPEF2 (Qu et al., 2023).
- CFAP73/MIA2 is a component of the MIA complex positioned between IDAF/11 and N-DRC (Yamamoto et al., 2023); MIA complex components, FAP73 and FAP100, together with linker-forming FAP57, were mistakenly listed as IDAF/11 in Walton et al., 2023;
- DLEC1, also named FAP81, is a component of C1a-e-c supercomplex (Fu et al., 2019).
- CCDC13 listed as a protein with "unknown localization" was recently described as essential for CP formation, directly interacting with Spef1 in *Drosophila* and being present along the entire cilia length (Wu et al., 2025).

Minor comments

Introduction

1. Page 3, line 45: "...motile cilia occur in epithelial cells..." - "occur" is not the best word; please rephrase.
2. Page 3, line 46: "in sperm cells...as monocilia (one cilium per cell), or flagellum." Change to: "named flagellum"
3. Page 3, line 54: "...are decorated with axonemal dyneins and radial spokes." Suggestion: please add: "...are decorated with multiprotein complexes, including axonemal dyneins and radial spokes."
4. Page 3, line 58 - the interactions between RS and CP are not only mechanical; please also cite Grossman-Haham et al., 2021).
5. Page 4, lines 66-72: the authors stated that proteins "critical for CP-MT formation in protists do not appear to exhibit similar roles in mammals." This statement is not quite true. In both protists [*Chlamydomonas* (Dymek et al., 2004, Dymek and Smith, 2012) and *Tetrahymena* (Sharma et al., 2007)] katanin mutants assemble CP-less cilia/flagella, suggesting that also in protists katanin likely plays a role in CP seed formation.
In the case of PF16, in mammals, lack of PF16 caused misorientation of CP in ependymal and respiratory cilia (Teves et al., 2014), similar to *Trypanosoma* (Ralston et al., 2006), while in *Chlamydomonas* (Smith and Lefebvre, 1996) and *Plasmodium* microgamete (Straschil et al., 2010) PF16 mutant flagella lack either the entire PC complex or one of the CP microtubules.

Results

6. Page 7, lines 143-145 - the presence of some of the listed proteins in CP projections was earlier shown using genetic and biochemical methods. Please cite those papers.
7. Page 8: please, also cite an earlier paper showing that RSP/RSPH4 is an RS head protein.
8. Page 8, lines 156-157: Since a paper describing a role of CCDC13 is already published, please add that besides JHY, also CCDC13 is implicated in CP-MT formation.
9. Page 9, line 189: the authors stated that: "GFP-CCDC13 exhibited a distribution similar to that of RSPH4". Based on the provided images, I cannot agree with the authors. The RSP4A signal looks like a hollow tube that agrees with the presence of RS on the outer doublets (RSP4A is an RS head protein) but not in the cilium lumen. In contrast, the CCDC13 signal "fills" the entire cilium lumen, so it is present also in the region of the CP, although it occupies a broader area than other analyzed CP proteins.

Figures

Fig 1C - GO analyses; please add in the Fig description a number of proteins belonging to other GO and thus not shown in Fig1C.

Fig 1E- CFAP73/MIA2 is a component of the MIA complex (Yamamoto et al., 2013), not IDA; ADGB (androglobin) is likely a CP C1b projection protein (please see above).

Response letter

Referee #1:

This manuscript presents a comprehensive study on the functional significance of the Juvenile hydrocephalus (JHY) protein. While previous research identified the *jhy* gene as a key factor leading to rapidly progressive hydrocephalus and revealed structural defects in cilia due to its absence, the precise structural and biological roles of the JHY protein itself have remained unclear.

The authors convincingly demonstrate that JHY is essential for the formation of the central apparatus (CA) in functional motile cilia of multiciliated mouse endothelial progenitor cells (mEPCs). The conclusions are supported by a robust and multi-faceted experimental approach, including:

1. Proximity proteomics using WDR47, a central regulator of central pair microtubule (CP-MT) formation and maintenance, as bait.
2. Subcellular localization studies-using both high-resolution fluorescence microscopy and antibody-based detection-to track JHY and CP-MT-associated proteins.
3. Co-immunoprecipitation assays employing GFP-tagged, FLAG-tagged, and deletion mutants to validate the interaction between JHY and WDR47.
4. Gene expression analysis of *jhy* across different tissues via qPCR.
5. Phenotypic characterization of *jhy* knockout mice, including quantification of basal bodies and cilia per cell, assessment of ciliary beat frequency, and immunohistochemical analysis of HYDIN, WDR47, and SPEF1 in mEPCs.
6. Phylogenetic analysis of *jhy* orthologs to explore evolutionary conservation.

I am highly impressed by the quality and clarity of the experimental work. The data are exceptionally clean and the logical flow of the experimental design and result interpretation is outstanding-reminiscent of textbook-level clarity, which is rarely encountered in peer-reviewed articles.

The manuscript is well written, the English is easy to follow, and the figures are carefully constructed with informative legends.

One potential next step for the authors would be to perform a high-resolution structural analysis of the JHY protein, such as through cryo-electron microscopy or X-ray crystallography. Such structural insights could further elucidate the molecular mechanisms by which JHY contributes to central apparatus formation and function. However, I recognize that this would be a substantial undertaking beyond the scope of the current study and is not necessary for publication in EMBO Reports. The current dataset is sufficiently complete and compelling to warrant publication as is.

Response: We sincerely thank the reviewer for the time and effort in evaluating our work and appreciate the positive comments. We agree with the reviewer that the high-resolution structure of JHY is important, but can be left to future studies.

One very minor point: while it is evident that *jhy* refers to Juvenile hydrocephalus, the authors should define this abbreviation explicitly at its first appearance in the main text. Aside from this ignorable minor correction, I strongly recommend publication of this manuscript in EMBO Reports without further revision.

Response: We thank the reviewer for the suggestion and have revised the manuscript to include this information.

Referee #2:

EMBOR-2025-62146-T

"JHY enables the transition from switchable to fixed ciliary waveforms in metazoan evolution"

Summary

- 1. Does this manuscript report a single key finding?

YES

JHY interacts with SPEF1 and WDR47 and plays a role in central pair microtubule stabilization in mouse ependymal cilia.

- 2. Is the reported work of significance (YES), or does it describe a confirmatory finding or one that has already been documented using other methods or in other organisms etc?

YES

The significance of JHY in CP-MT assembly/stability was not earlier investigated.

- 3. Is it of general interest to the molecular biology community?

YES

The mechanism(s) enabling non-centrosomal CP-MT formation/stability, indispensable for proper functioning of motile cilia, is poorly understood despite several recently published papers.

- 4. Is the single major finding robustly documented using independent lines of experimental evidence (YES), or is it really just a preliminary report requiring significant further data to become convincing, and thus more suited to a longer format article (NO)?

YES

While searching for proteins involved in CP-MT assembly, the authors expressed WDR47, a key CP assembly factor, in fusion with APEX2 and performed a proximity assay. The comparison of the putative Wdr47 interactome encompassing 2448 proteins with Wdr47 knockout ciliomes led to the identification of 43 candidate proteins both enriched in the WDR47 interactome and significantly reduced in KO. After localization studies of some of the identified CP components or proteins of unknown intraciliary position, the authors focused their interest on JHY, a protein earlier shown to be linked to the CP-MT formation.

Using microscopic and biochemical approaches, the authors showed that JHY directly interacts with WDR47 and SPEF1, a protein known to bind to MT seam to stabilize MTs, and that these proteins co-localize in mouse ependymal cilia in a cilia-length-dependent manner. The JHY knockout mice exhibited PCD-related phenotype, including hydrocephalus. Cilia assembled by Jhy KO mEPCs displayed rotatory movement and a CP-loss, especially in the distal part of the cilia. Based on collected data, the authors concluded that SPEF1 recruits JHY and WDR47 to central MTs and that JHY plays a role in CP-MT stabilization.

Identification of JHY as a protein that directly interacts with WDR47 and SPEF1 is a new and interesting finding, significantly contributing to our understanding of the molecular mechanism of CP-MT assembly and stability. The provided biochemical and microscopic data are convincing and surely will be of interest to all who study motile cilia or cilia-related diseases.

Response: We sincerely thank the reviewer for the positive comments and for helping us to substantially improve the manuscript.

However, there are also some weak points in this manuscript. The manuscript can be shortened and perhaps the order of the chapters reorganized. The last chapter describing phylogenetic analyses should be shorter, more concise, and perhaps some parts should be combined with the relevant discussion section.

Response: We thank the reviewer for the suggestion. Following the request, we have modified the last section and related discussion in the revised manuscript to make them more concise.

The authors suggested that obtained proximity proteomics data can be used to identify components of CA projection. However, the putative WDR47 interactome containing 2448 proteins (only 14% are ciliary proteins) is not sufficient to select such candidates. Even among ciliary proteins, one can find numerous doublet proteins. If the JHY was not earlier shown to be related to CP formation, would the authors select JHY for further analyses based exclusively on WDR47 interactome and the WDR47 KO data?

Response: We appreciate these comments. We agree with our reviewer that the use of the proximity proteomics data for the identification of CA-projection components is not straightforward. We thus have deleted the sentence “In addition, the proximity proteome can also be used to identify CA projection components unique to mammals or even metazoans” from the Discussion section in the revised manuscript.

As we aim to identify proteins critical for the CP-MT formation, we chose the strategy in Figure 1D to efficiently narrow down to the pool of candidate proteins. As this pool contains all the known CP-MT regulators and is also abundant in CA components (Fig. 1E), we reasoned that it was suitable for a localization screen.

In the screen, we actually identified more CP localized proteins than those presented in Figure 2. As time went on, however, many proteins became documented in the literature, though these publications also helped us narrow down our candidates. Regarding JHY, our findings that it is a WDR47-related, CP-MT-associated protein (Figs. 1-3) and previous publications on JHY-deficient mice are both important reasons for choosing it for further studies. In fact, JHY is not the only candidate that we are interested in. The function of CCDC13 (Fig. 2B) in the CP-MT formation, for instance, is recently reported through joint efforts involving some of us (Wu et al., 2025). We have updated the information on CCDC13 in the revised manuscript.

Major comments:

Controls for the APEX2 experiment (WT cilia, cilia from cells expressing only APEX2) showing a background biotinylation are missing. It is surprising that only 14% of the proteins identified in this experiment were annotated as ciliary proteins. Perhaps this is also due to the identification of some unspecifically biotinylated proteins. Did the authors try to optimize assay conditions? Without additional data, the presented interactome is rather insufficient to identify remaining CA projection components as the authors suggest at the end of the Discussion.

Response: In the proximity ligation experiments, our negative control was mock-treated (control) mEPCs expressing APEX2-WDR47. Following the request, we have included control MS results in Datasets EV1 and accordingly updated Figure 1C-E and Dataset EV4 in the revised manuscript. The filtered proximity proteomics data (peptide ratio ≥ 2) contain 1,923 proteins (Fig. 1D; Dataset EV1). The Venn overlap analysis yields 42 hits (Dataset EV4; Fig. 1D,E). All these proteins are found in the previous pool of 43 proteins that additionally contains SPAT6, a protein of unknown localization. Therefore, the lack of SPAT6 in the revised manuscript does not influence the conclusion of this part of the results.

As expected, the percentage of ciliary proteins increases (21%) in the set of filtered proximity proteomics data. The major reason for such a relatively low percentage is largely due to our use of whole cell lysates to purify biotinylated proteins for MS (please refer to Methods, under subtitle “Proximity labeling assay”). As APEX2-WDR47-induced biotinylated proteins were highly enriched in cilia (Fig. 1B), we reasoned that cilia purification, which generally requires many more mEPCs, was not necessary. Furthermore, as some WDR47 interactors in cilia may interact with APEX2-WDR47 in cytosols as well, whole cell lysates might contain additional amounts of biotinylated WDR47 interactors. The relatively high abundance of CAMSAPS, JHY, and SPEF1 in the MS results (Dataset EV1) supports the effectiveness of the strategy. As we responded previously, we agree with our reviewer that the proximity proteomics data alone are insufficient for the identification of unknown CA projection components and have deleted the sentence “In addition, the proximity proteome can also be used to identify CA projection components unique to mammals or even metazoans” from the Discussion section in the revised manuscript.

While in the Dataset EV1, the identified proteins are ordered according to the decreasing number of peptides, the reason for ordering proteins in Dataset EV4 as they are ordered is unclear (proteins are neither in alphabetical order (protein names or accession number) nor according to the number of peptides).

Response: We apologize for this and have reformatted Dataset EV4 so that proteins are arranged in order of decreasing number of peptides in the APEX2-WDR47 sample.

I understand that it is not easy, but can the authors next include graphs showing changes in the intensity of IF signals along the cilia length to support their model of protein localization during cilia assembly and elongation?

Response: Following the suggestion, we attempted to plot fluorescent intensities along individual cilia, and examples are presented below (Fig. 1 for reviewers). Overall, the plots align with our observations. However, a major issue is the difficulty in identifying well-separated cilia for plotting because fluorescent signals from overlapping cilia can lead to misinterpretation of the intensity curves. The punctate nature of the ciliary fluorescent signals further complicates this task. Therefore, we decided not to include such graphs in the revised manuscript.

Fig. 1 for reviewers

Missing REFs:

- page 7: Studies in a model organism, Tetrahymena, showed that ADGB is a subunit of C1b projection (Joachimiak et al., 2021); those findings were supported by Qu et al., showing direct interactions between ADGB and two C1b proteins, CFAP69 and SPEF2 (Qu et al., 2023).
- CFAP73/MIA2 is a component of the MIA complex positioned between IDAF/I1 and N-DRC (Yamamoto et al., 2023); MIA complex components, FAP73 and FAP100, together with linker-forming FAP57, were mistakenly listed as IDAF/I1 in Walton et al., 2023;
- DLEC1, also named FAP81, is a component of C1a-e-c supercomplex (Fu et al., 2019).
- CCDC13 listed as a protein with "unknown localization" was recently described as essential for CP formation, directly interacting with Spef1 in Drosophila and being present along the entire cilia length (Wu et al., 2025).

Response: We appreciate the reviewer's suggestions. In the revised manuscript, we have included these references in the text and updated the related content in Fig. 1E.

Minor comments

Introduction

1. Page 3, line 45: "...motile cilia occur in epithelial cells..." - "occur" is not the best word; please rephrase.

Response: We thank the reviewer for pointing this out and have replaced "occur" with "are distributed" in the revised manuscript.

2. Page 3, line 46: "in sperm cells...as monocilia (one cilium per cell), or flagellum." Change to: "named flagellum"

Response: We have modified the text accordingly in the revised manuscript.

3. Page 3, line 54: "...are decorated with axonemal dyneins and radial spokes." Suggestion: please add: "...are decorated with multiprotein complexes, including axonemal dyneins and radial spokes."

Response: We have modified the text accordingly in the revised manuscript.

4. Page 3, line 58 - the interactions between RS and CP are not only mechanical; please also cite Grossman-Haham et al., 2021).

Response: We have included this reference in the text and modified the text in the revised manuscript.

5. Page 4, lines 66-72: the authors stated that proteins "critical for CP-MT formation in protists do not appear to exhibit similar roles in mammals." This statement is not quite true. In both protists [*Chlamydomonas* (Dymek et al., 2004, Dymek and Smith, 2012) and *Tetrahymena* (Sharma et al., 2007)] katanin mutants assemble CP-less cilia/flagella, suggesting that also in protists katanin likely plays a role in CP seed formation.

In the case of PF16, in mammals, lack of PF16 caused misorientation of CP in ependymal and respiratory cilia (Teves et al., 2014), similar to *Trypanosoma* (Ralston et al., 2006), while in *Chlamydomonas* (Smith and Lefebvre, 1996) and *Plasmodium* microgamete (Straschil et al., 2010) PF16 mutant flagella lack either the entire PC complex or one of the CP microtubules.

Response: We appreciate these comments. The entire sentence is "CA proteins identified as critical for CP-MT formation in protists do not appear to exhibit similar roles in mammals". While we agree with our reviewer that katanin is a conserved player in the CP formation, neither PF15p nor PF19p, the regulatory (p80) and catalytic (p60) subunits of Katanin in protists, display CP localization (Dymek et al., 2004; Dymek and Smith, 2012; Sharma et al., 2007). Katanin is thus not a CA protein.

On the other hand, although PF16p and PF20p are CA proteins and their mutations in *Chlamydomonas* tend to result in 9+0 and 9+1 axonemes (Adams et al., 1981; Dutcher et al., 1984; Smith and Lefebvre, 1996; Smith and Lefebvre, 1997), mice deficient in *Spag6* (*pf16* ortholog), *Spag16* (*pf20* ortholog), and both *Spag6* and *Spag16* display normal (9+2) axonemes (Teves et al., 2014; Zhang et al., 2006; Zhang et al.,

2007). SPAG6 and SPAG16 are thus dispensable for the CP-MT formation. We found that we missed the publication by Teves and colleagues (Teves et al., 2014) and have cited this paper in the revised manuscript.

Results

6. Page 7, lines 143-145 - the presence of some of the listed proteins in CP projections was earlier shown using genetic and biochemical methods. Please cite those papers.

Response: We thank our reviewer for pointing this out and have updated the references in the revised manuscript.

7. Page 8: please, also cite an earlier paper showing that RSP/RSPH4 is an RS head protein.

Response: Following the request, we have included two earlier papers in the revised manuscript.

8. Page 8, lines 156-157: Since a paper describing a role of CCDC13 is already published, please add that besides JHY, also CCDC13 is implicated in CP-MT formation.

Response: We have added this reference and revised the text accordingly in the manuscript (page 7, line 131-132).

9. Page 9, line 189: the authors stated that: "GFP-CCDC13 exhibited a distribution similar to that of RSPH4". Based on the provided images, I cannot agree with the authors. The RSP4A signal looks like a hollow tube that agrees with the presence of RS on the outer doublets (RSP4A is an RS head protein) but not in the cilium lumen. In contrast, the CCDC13 signal "fills" the entire cilium lumen, so it is present also in the region of the CP, although it occupies a broader area than other analyzed CP proteins.

Response: We appreciate the comments. In the revised manuscript, we have rephrased this sentence to better describe our results.

Figures

Fig 1C - GO analyses; please add in the Fig description a number of proteins belonging to other GO and thus not shown in Fig1C.

Response: We appreciate the reviewer's suggestion. In the revised manuscript, we have provided this information in the relevant figure legend.

Fig 1E- CFAP73/MIA2 is a component of the MIA complex (Yamamoto et al., 2013), not IDA; ADGB (androglobin) is likely a CP C1b projection protein (please see above).

Response: We apologize for this oversight and have updated the figure and related text in the revised manuscript.

References:

- Adams, G.M., B. Huang, G. Piperno, and D.J. Luck. 1981. Central-pair microtubular complex of *Chlamydomonas* flagella: polypeptide composition as revealed by analysis of mutants. *J Cell Biol.* 91:69-76.
- Dutcher, S.K., B. Huang, and D.J. Luck. 1984. Genetic dissection of the central pair microtubules of the flagella of *Chlamydomonas reinhardtii*. *J Cell Biol.* 98:229-236.
- Dymek, E.E., P.A. Lefebvre, and E.F. Smith. 2004. PF15p is the *Chlamydomonas* homologue of the Katanin p80 subunit and is required for assembly of flagellar central microtubules. *Eukaryot Cell.* 3:870-879.
- Dymek, E.E., and E.F. Smith. 2012. PF19 encodes the p60 catalytic subunit of katanin and is required for assembly of the flagellar central apparatus in *Chlamydomonas*. *J Cell Sci.* 125:3357-3366.
- Sharma, N., J. Bryant, D. Wloga, R. Donaldson, R.C. Davis, M. Jerka-Dziadosz, and J. Gaertig. 2007. Katanin regulates dynamics of microtubules and biogenesis of motile cilia. *J Cell Biol.* 178:1065-1079.
- Smith, E.F., and P.A. Lefebvre. 1996. PF16 encodes a protein with armadillo repeats and localizes to a single microtubule of the central apparatus in *Chlamydomonas* flagella. *J Cell Biol.* 132:359-370.
- Smith, E.F., and P.A. Lefebvre. 1997. PF20 gene product contains WD repeats and localizes to the intermicrotubule bridges in *Chlamydomonas* flagella. *Mol Biol Cell.* 8:455-467.
- Teves, M.E., P.R. Sears, W. Li, Z. Zhang, W. Tang, L. van Reesema, R.M. Costanzo, C.W. Davis, M.R. Knowles, J.F. Strauss, 3rd, and Z. Zhang. 2014. Sperm-associated antigen 6 (SPAG6) deficiency and defects in ciliogenesis and cilia function: polarity, density, and beat. *PLoS One.* 9:e107271.
- Wu, Z., Y. Zhang, J. Liu, H. Liu, J. Niu, Y. Li, S. Xie, X. Yan, X. Zhu, and Q. Wei. 2025. Ccdc13 is essential for the assembly of ciliary central microtubules. *Natl Sci Rev.* 12:nwaf095.
- Zhang, Z., I. Kostetskii, W. Tang, L. Haig-Ladewig, R. Sapiro, Z. Wei, A.M. Patel, J. Bennett, G.L. Gerton, S.B. Moss, G.L. Radice, and J.F. Strauss, 3rd. 2006. Deficiency of SPAG16L causes male infertility associated with impaired sperm motility. *Biol Reprod.* 74:751-759.
- Zhang, Z., W. Tang, R. Zhou, X. Shen, Z. Wei, A.M. Patel, J.T. Povolishock, J. Bennett, and J.F. Strauss, 3rd. 2007. Accelerated mortality from hydrocephalus and pneumonia in mice with a combined deficiency of SPAG6 and SPAG16L reveals a functional interrelationship between the two central apparatus proteins. *Cell motility and the cytoskeleton.* 64:360-376.

Dear Prof. Zhao,

Thank you for submitting your revised manuscript. It has now been seen by one of the original referees.

As you will see, the referee finds that the study is significantly improved during revision and recommends publication. However, the referee has a few minor remaining concerns: they request a clarification on the number of candidate proteins at one point in the text and suggest the mention of another protein at another point. Please address all concerns textually per referee recommendations. Please provide a point-by-point response. Please let me know if you would like to discuss any of the points further.

Additionally, I have adjusted your abstract to ensure that it is in present tense. Please see the adjusted abstract below my sign off for your consideration.

Moreover, the editorial points below need to be addressed before I can accept the manuscript.

- Please reduce the number of keywords on the abstract page to five (ideally choosing broad general terms).
- Please note that the data availability statement should only refer to data deposited in publicly available repositories. Therefore, please remove the statement "Materials generated in this study are available from the corresponding author upon reasonable request."
- Please move the Disclosure and Competing Interest Statement to after the Acknowledgements.
- CRediT has replaced the traditional author contributions section because it offers a systematic machine readable author contributions format that allows for more effective research assessment. Please remove the Author Contributions section from the manuscript and use the free text boxes beneath each contributing author's name in our online systems to add specific details on the author's contribution. More information is available in our guide to authors.
- Please note that there is a pop-up message regarding formulas that appears when the Author Checklist is opened. This should be rectified.
- Please note that EMBO press papers are accompanied online by:
 - A) a short (2 sentences) summary of the findings and their significance,
 - B) 2-5 short bullet points highlighting the key results, and
 - C) a synopsis image in .jpg or .png format that is exactly 550 pixels wide and 300-600 pixels high (the height is variable). Please note that the text needs to be legible at the final size. Please upload this information along with your revised manuscript (the text for A and B should be provided in one separate Word file uploaded as Synopsis text).
- Please download and fill our Reagents and Tools Table template (.docx), which you can find in our author guidelines: <https://www.embopress.org/page/journal/14693178/authorguide#structuredmethods>. When submitting your revised manuscript, please do not include the Reagents and Tools Table in the Methods section of the manuscript but upload it as a separate file choosing the file type "Reagent Table".
- Please provide explicit statements on blinding and randomization used to quantify data, on animal housing, and on cell authentication, as indicated in the Author checklist.
- Please update the citations for BioRxiv preprints. The form of the citation in the text is: (preprint: NAME1 et al, YEAR); in the reference list: Author NAME1, Author NAME2 (YEAR) article title. bioRxiv doi [PREPRINT].
- Previously-published datasets re-analyzed in the current manuscript should be cited. Example:
References
Hörnberg E, Ylitalo EB, Crnalic S, Antti H, Stattin P, Widmark A, Bergh A, Wikström P (2011) Gene Expression Omnibus GSE29650 (<https://www.ncbi.nlm.nih.gov/geo/query/acc.cgi?acc=GSE29650>). [DATASET]
Hörnberg E, Ylitalo EB, Crnalic S, Antti H, Stattin P, Widmark A, Bergh A, Wikström P (2011) Expression of androgen receptor splice variants in prostate cancer bone metastases is associated with castration-resistance and short survival. PLoS One 6: e19059
- Please provide the raw values for figure 4A's source data.
- Please upload Figure EV1 as a Figure file similar to the main figures (its legend should stay in the manuscript file).
- Please upload Datasets EV1-EV4 as separate datasets (file type Data set), and the spreadsheets each need a separate "Legend" tab containing dataset title and legend information, that should be removed from the manuscript file.

- Please upload Tables EV1 and EV2 as Expanded View Content individual files. The legends of EV tables need to be removed from the manuscript file.
- Please zip Movie EV1 with a separate text file containing its legend and upload as zip folder Movie EV1. The legend needs to be removed from the manuscript file.
- Source data files need to be saved in a scheme one figure/folder and then uploaded as .zip files. E.g. all the Source data files for figure 1 need to be saved in a single folder and this needs to be zipped and then uploaded as "SD figure 1.zip" file. For EV and/or appendix figures, ZIP together all source data.
- Scale bar lengths (e.g. in Figure 1B) should be moved to the figure legends.

Our production/data editors have asked you to clarify several points in the figure legends:

- Please note that the exact p values are not provided in the legend of figure 5E, 6B. This needs to be rectified.
- Please indicate the statistical test used for data analysis in the legend of figure 1C
- Please note that the asterisk is not defined in the legend of figure 4H. This needs to be rectified.

Kind regards,
Kurt Weir
Editor
EMBO Reports

"Motile cilia are evolutionarily conserved protrusions critical for motility and homeostasis. Their rhythmic movements require the central pair of microtubules (CP-MTs). While the initial CP-MT assembly in mammals is mediated by WDR47 and microtubule minus-end-binding CAMSAPs, the mechanism by which CP-MTs are stabilized remains unclear. Here, we demonstrate that WDR47 coordinates JHY and SPEF1 to maintain the stability of mammalian CP-MTs. By generating a proximity interactome of WDR47, we identify a group of CP-MT-associated proteins, including SPEF1 and JHY. WDR47 enriches JHY and SPEF1 to the central lumen and tip of nascent cilia, whereas SPEF1 recruits WDR47 and JHY to CP-MTs through direct interactions. Jhy deficiency in mice preferentially disrupts distal CP-MTs, resulting in rotatory ciliary beats. Phylogenetic analyses suggest conserved functions of WDR47 and SPEF1 in protozoa and metazoans, as well as a role for JHY in animals with radial or bilateral body symmetry. We propose that JHY emerges to further reinforce CP-MTs, enabling the transition from switchable to fixed ciliary waveforms in metazoan evolution."

Referee #2:

I have only two very minor remarks that can be corrected at any time:

1. Page 7, line 131: "identified 42 common proteins", Fig 1D - also 42. Please check Fig. 1 description: "43 candidate proteins", also page 17, line 381 - 43 candidates
2. Page 7, line 137: CFAP69 and likely CFAP246/LRGUK are C1b projection proteins (Joachimciak et al., 2021). CFAP69 should be listed together with SPEF2 (ii) and perhaps also CFAP246; CFAP246 is similar to the N-terminal part of human LRGUK, but in both *Chlamydomonas* and *Tetrahymena* lacks the GK domain

This is a very interesting research, and it was a pleasure to review it. I would like to thank the Editor for including me in the revision process, and I congratulate the Authors on this work.

Response letter

Referee #2:

I have only two very minor remarks that can be corrected at any time:

1. Page 7, line 131: "identified 42 common proteins", Fig 1D - also 42. Please check Fig. 1 description: "43 candidate proteins", also page 17, line 381 - 43 candidates

Response: We apologize for this oversight and have corrected the error in the revised manuscript.

2. Page 7, line 137: CFAP69 and likely CFAP246/LRGUK are C1b projection proteins (Joachimciak et al., 2021). CFAP69 should be listed together with SPEF2 (ii) and perhaps also CFAP246; CFAP246 is similar to the N-terminal part of human LRGUK, but in both *Chlamydomonas* and *Tetrahymena* lacks the GK domain.

Response: We appreciate the reviewer's comments. The *Tetrahymena thermophila* SPEF2A, CFAP69, CFAP246/LRGUK, and ADGB have been localized to the C1b/C1f projection (Joachimciak et al., 2021). However, the exact localization and function of their mammalian homologs remain to be determined. To better describe this result, we have added this information to the revised manuscript.

This is a very interesting research, and it was a pleasure to review it. I would like to thank the Editor for including me in the revision process, and I congratulate the Authors on this work.

Response: We sincerely thank the reviewer for the positive comments and for helping us to improve the manuscript substantially.

Prof. Huijie Zhao
Shandong Normal University
College of Life Sciences
88 East Wenhua Road
Jinan, Shandong 250014
China

Dear Prof. Zhao,

I am pleased to inform you that your manuscript has been accepted for publication in EMBO reports. Your manuscript will be processed for publication by EMBO Press. It will be copy edited and you will receive page proofs prior to publication. Please note that you will be contacted by Springer Nature Author Services to complete licensing and payment information.

Yours sincerely,

Kurt Weir
Editor
EMBO Reports
